# Gibbs Sampling with Simulated Annealing K-Means for Mixture Regression

## Abstract

Fitting the Mixture of Multivariate Linear Regression models (MMLR) is a fundamental task in the analysis of heterogeneous data. Still, standard methods like the EM and K-means algorithms are hindered by their convergence to local optima and the NP-hard nature of the underlying optimization problem. To address this fundamental challenge, we propose Gibbs sampling with the simulated annealing K-means clustering algorithm. By synergizing the K-means framework with Gibbs sampling and a simulated annealing schedule, this approach is provably robust to initialization and avoids poor local minima. The primary contributions of this work are a comprehensive set of theoretical guarantees. First, we provide the first non-asymptotic guarantees on the algorithm's convergence to the global minimum of the Within-Cluster Sum of Squares (WCSS) objective, establishing explicit bounds on its rate and probability of convergence. Second, based on this global optimum, we establish a rigorous upper bound for the estimation error of the regression coefficients and a lower bound on classification accuracy in an asymptotic sense. Numerical experiments validate the superior performance of our method. This work presents a theoretically grounded and computationally practical framework for estimation and clustering in mixture regression models.

## 1 Introduction

Understanding the linear relationships between sets of high-dimensional variables is a fundamental goal in numerous scientific and industrial domains (James et al., 2013). Multivariate Linear Regression (MLR) serves as the cornerstone of this task, modeling how multiple predictor variables jointly influence numerous predicted variables (Härdle & Simar, 2007; Hastie, 2009). However, a key limitation of standard MLR is its assumption of data homogeneity, presupposing that a single regression model can adequately describe the entire data set (Goldfeld & Quandt, 1973; Jacobs et al., 1991; McLachlan et al., 2019). In practice, many datasets exhibit significant heterogeneity, comprising several latent subgroups with distinct relational patterns (Hennig et al., 2015; McLachlan et al., 2019). For example, in personalized medicine, different subpopulations of patients may respond to treatments in unique ways (Hamburg & Collins, 2010; Collins & Varmus, 2015; Shen & He, 2015). A mixture of Multivariate Linear Regression (MMLR) models provides an elegant solution to this challenge, capable of simultaneously clustering data into coherent groups and fitting a tailored MLR model to each, thus capturing the underlying heterogeneous structure (De Veaux, 1989; Jacobs et al., 1991; Frühwirth-Schnatter, 2006; McLachlan et al., 2019).

For decades, the Expectation-Maximization (EM) algorithm (Dempster et al., 1977) and K-means algorithm (Lloyd, 1982) have been the workhorse methods for fitting these models, prized for their simplicity and computational efficiency. Despite their widespread success, classical algorithms like EM and K-means are hindered by their iterative and local optimization nature. They are only guaranteed to converge to a local optimum. They are thus susceptible to parameter initialization, which has significantly limited the reliability of mixture models in critical applications (McLachlan et al., 2019). To mitigate this, a variety of practical strategies have been developed, such as K-means++ to obtain better starting points (Arthur & Vassilvitskii, 2007) and multiple random restarts (Jain, 2010).

However, these practical methods lack theoretical guarantees and the establishment of such properties is exceptionally difficult. Previous theoretical work often relied on the impractical technique of "sample splitting" to make the analysis tractable (Yi & Caramanis, 2015; Zhang et al., 2020).

Pioneering work (Wang et al., 2024) has made a significant breakthrough by establishing a rigorous convergence rate analysis for a penalized EM algorithm in high-dimensional mixture linear regression without sample splitting. However, this theoretical guarantee has its own limitations as it is based on the strong assumption that the algorithm must be initialized within a "contraction basin" close to the true parameters. In fact, despite these extensive efforts, finding the global optimum for mixture models remains an NP-hard problem (Aloise et al., 2009), highlighting the need for fundamentally new approaches.

To address this challenge fundamentally, we introduce a novel Gibbs sampling with the simulated annealing K-means clustering algorithm. Our approach synergizes the efficiency of the K-means framework with the global exploration capabilities of stochastic optimization. We augment the classic assignment-update loop with a Gibbs sampling step to probabilistically explore cluster assignments (Geman & Geman, 1984) and a simulated annealing (SA) schedule to escape poor local minima (Kirkpatrick et al., 1983; Klein & Dubes, 1989). The efficacy of SA is grounded in solid theory; for instance, recent work by Tang & Zhou (2021) provides a rigorous convergence analysis, proving that the probability that the algorithm remains far from the global optimum exhibits a polynomial decay over time. This quantitative support justifies its ability to guide the search globally. By integrating these powerful stochastic techniques, which have been shown to improve deterministic methods (Selim & Alsultan, 1991), our hybrid design ensures robustness to initialization and facilitates convergence towards a globally optimal solution.

To validate these claims, our algorithm was validated through extensive experiments on simulated datasets, where it consistently outperformed standard baselines. Beyond this empirical result, our primary contribution is a comprehensive set of theoretical guarantees for this method. In sharp contrast to the well-documented local convergence properties of traditional methods such as EM and K-means (Balakrishnan et al., 2017; McLachlan et al., 2019), we prove that our algorithm is robust to initialization and converges to the global optimum.

Our work establishes the first non-asymptotic (finite-sample) guarantees on its convergence rate and probability of convergence. This type of analysis aligns with a modern push in stochastic optimization to provide explicit performance bounds, similar to recent advances in the theoretical understanding of core components such as simulated annealing (Tang & Zhou, 2021). Furthermore, based on the global optimality guaranteed by our algorithm, we analyze the statistical properties of the resulting estimator. We establish rigorous upper limits on the estimation error of the regression coefficients in both asymptotic and non-asymptotic regimes, complementing previous work on penalized estimators for mixture models (Städler et al., 2010; Wang et al., 2024). Finally, we derive a formal lower bound on the accuracy of the algorithm's classification in both asymptotic and non-asymptotic (finite-sample) regimes, addressing the inherent difficulty of recovering latent labels in mixture models (Von Luxburg, 2007).

## 2 MODEL AND ALGORITHM

We denote $X \in \mathbb{R}^p$ as predictors and $Y \in \mathbb{R}^q$ as the predicted variables from a mixture multivariate regression model. The mixing weight of the $k$th submodel is $p_k$, where $1 \le k \le K$. Then our model is formulated as,

$$Y = XB_U + \epsilon \quad s.t. \ X \sim N(0, \Sigma), \quad \epsilon \sim N(0, \sigma^2 I_q), \tag{1}$$
$$p(U = k) = p_k, \qquad B_{k,0} \in \mathbb{R}^{p \times q}, \qquad X \perp u$$

We consider a set of $n$ i.i.d. samples $\mathcal{S} = \{(x_i, y_i, u_{i,0})\}_{i=1}^n$ generated from the mixture model in (1), where the true cluster assignments $\mathcal{U}_0 = \{u_{1,0}, \cdots, u_{n,0}\}$ are not observed. Our objective is to estimate the true regression parameters $\theta_0 = \{B_{1,0}, \cdots, B_{K,0}\}$ with an estimator $\hat{\theta} = \{\hat{B}_1, \cdots, \hat{B}_K\}$. This, in turn, allows us to infer the cluster labels for the given samples as $\hat{\mathcal{U}} = \{\hat{u}_1, \cdots, \hat{u}_n\}$ and for new observations.

To develop our method, we first generalize the K-means approach to multivariate linear regression. Specifically, we define the *Within-Cluster Sum of Squares (WCSS)* function as:

$$J(\theta, \mathcal{U}) = \sum_{k=1}^K \sum_{u_i = k} \|y_i - x_i B_k\|_2^2. \tag{2}$$

According to a standard property of the K-means algorithm, for any $1 \leq i \leq n$, we have the estimate clusters $\hat{u}_i = \arg\min_k \|y_i - x_i \hat{B}_k\|_2^2$. We define the residual squares associated with the estimator $\theta$ as the function $m$:

$$m(x, y, \theta) = \min_{1 \leq k \leq K} \|y - xB_k\|_2^2.$$

Then for the $K$-means estimator $\hat{\theta}, \hat{\mathcal{U}}$ we have $\frac{J(\hat{\theta}, \hat{\mathcal{U}})}{n} = \frac{1}{n} \sum_{i=1}^n m(x_i, y_i, \hat{\theta})$. Similarly to traditional $K$-means clustering in Euclidean space, finding the global minimum of WCSS objective function is widely recognized as an NP-hard problem (Aloise et al., 2009). Consequently, deterministic algorithms that guarantee convergence to the global minimum are deemed intractable for practical purposes. We therefore propose a probabilistic framework that combines Gibbs sampling with simulated annealing. This approach ensures asymptotic convergence to the global minimum with high probability, provided that the computational complexity is bounded.

Within the simulated annealing framework, let $T$ be the temperature parameter. The energy function $\mathcal{E}(\theta, \mathcal{U}, T)$ is defined as:

$$\mathcal{E}(\theta, \mathcal{U}, T) = \exp\left(-\frac{J(\theta, \mathcal{U})}{T}\right) = \prod_{k=1}^K \prod_{\substack{i \\ \hat{u}_i = k}} \exp\left(-\frac{\|y_i - x_i B_k\|_2^2}{T}\right) \quad (3)$$

For the energy function (3), we use an alternating Gibbs sampling scheme between the parameters $\theta$ and the estimated cluster assignments $\mathcal{U}$. The conditional distribution for each assignment variable $\hat{u}_i$ follows a categorical distribution, with probabilities proportional to

$$P(\hat{u}_i = k) \propto \exp\left(-\frac{\|y_i - x_i \hat{B}_k\|_2^2}{T}\right).$$

This distribution exhibits conditional independence given the parameters $\theta$, which means that the sampling of $\hat{u}_i$ depends only on the current parameter estimates $\hat{B}_k$.

For the regression coefficients $\hat{B}_k$, we define the design matrix $X_k^{\mathcal{U}}$ as the matrix formed by stacking the predictor vectors $x_i$ for all observations with $\hat{u}_i = k$, and $Y_k^{\mathcal{U}}$ as the corresponding response vector. The conditional posterior distribution for $\hat{B}_k$ derived from the energy function is:

$$p(\hat{B}_k) \propto \exp\left(-\frac{\|Y_k^{\mathcal{U}} - X_k^{\mathcal{U}} \hat{B}_k\|_2^2}{T}\right),$$

which corresponds to a matrix-normal distribution under appropriate priors.

However, when the design matrix $X_k^{\mathcal{U}}$ is rank-deficient (that is, $\text{rank}(X_k^{\mathcal{U}}) < p$), the integral of the function $\exp\left(-\frac{\|Y_k^{\mathcal{U}} - X_k^{\mathcal{U}} \hat{B}_k\|_2^2}{T}\right)$ in the parameter space $\hat{B}_k \in \mathbb{R}^{p \times q}$ diverges. To address this ill-posedness and allow adequate sampling of $\hat{B}_k$, we introduce a ridge regularization penalty to the energy function (3). This yields the modified energy function:

$$\mathcal{E}(\theta, \mathcal{U}, T) = \exp(-\frac{J(\theta, \mathcal{U})}{T}) \prod_{k=1}^K \exp(-\frac{\|B_k\|_2^2}{2\kappa}) = \prod_{k=1}^K \exp(-\frac{\|B_k\|_2^2}{2\kappa}) \prod_i \exp(-\frac{m(x_i, y_i, \theta)}{T}). \quad (4)$$

It follows that Gibbs sampling with the modified energy function (4) is equivalent to Bayesian inference under the following probabilistic model: the prior distribution for the vectorized regression coefficients $\text{vec}(\hat{B}_k)$ is Gaussian with $\text{vec}(\hat{B}_k) \sim \mathcal{N}(\mathbf{0}, \kappa \mathbf{I}_{pq})$, while the sampling model corresponds to equation (1) with $\sigma^2 = T/2$. Crucially, as the temperature $T \to 0$ during simulated annealing, the influence of the regularization term vanishes asymptotically. Consequently, the global minimizer of $\mathcal{E}(\theta, \mathcal{U}, T)$ converges to the minimizer of the WCSS objective:

$$\lim_{T \to 0} \arg\max_{\theta, \mathcal{U}} \mathcal{E}(\theta, \mathcal{U}, T) = \arg\min_{\theta, \mathcal{U}} J(\theta, \mathcal{U}) = \arg\min_{\theta, u_i = \arg\min_k \|y_i - x_i B_{k,0}\|} \sum_{i=1}^n m(x_i, y_i, \theta).$$

Under this Bayesian interpretation, the conditional distribution is as follows: For cluster assignments, the following distribution $P(\hat{u}_i = k) \propto \exp(-\frac{\|y_i - x_i \hat{B}_k\|_2^2}{T})$ is true, and the posterior distribution for vectorized coefficients is: $p(\text{vec}(\hat{B}_k)) \propto \exp\left(-\frac{\|Y_k^{\mathcal{U}} - X_k^{\mathcal{U}} \hat{B}_k\|_F^2}{T} - \frac{\|\hat{B}_k\|_F^2}{2\kappa}\right)$. This corresponds to a Gaussian distribution:

$$\text{vec}(\hat{B}_k) \sim \mathcal{N}\left(\text{vec}\left(\left(X_k^{\mathcal{U}\top} X_k^{\mathcal{U}} + \frac{T}{2\kappa} \mathbf{I}_p\right)^{-1} X_k^{\mathcal{U}\top} Y_k^{\mathcal{U}}\right), \frac{T}{2}\left(\mathbf{I}_q \otimes \left(X_k^{\mathcal{U}\top} X_k^{\mathcal{U}} + \frac{T}{2\kappa} \mathbf{I}_p\right)\right)^{-1}\right).$$

Based on the preceding discussion, we introduce our simulated annealing method, formally presented in Algorithm 1. The algorithm is designed to minimize the regularized energy function from Equation (4). A key component is its slow cooling schedule, where the temperature $T_t$ in iteration $t$ follows $T_t = T_0 \cdot \log(t+1)^{-\alpha}$ for a constant $0 < \alpha < 1$. Although our implementation employs a K-means++ seeding strategy (Arthur & Vassilvitskii, 2007) for practical efficiency, we prove in Section 3.3 that the algorithm is theoretically robust to the choice of initial parameters.

---

**Algorithm 1:** Gibbs sampling with simulated annealing K-means clustering algorithm for multivariate linear regression

---

**Input:** $[(x_1, y_1), (x_2, y_2), \cdots, (x_n, y_n)], K, \alpha, T_0, c, \kappa$
**Output:** $\hat{B}_1, \hat{B}_2, \cdots, \hat{B}_K, \hat{u}_1, \hat{u}_2, \cdots, \hat{u}_n$
Let $\hat{B}_1 = \hat{B}_2 = \cdots = \hat{B}_K = 0$ initially. **for** $k \leftarrow 1$ **to** $K$ **do**
    $r \leftarrow \{1, 2, \cdots, n\}$ satisfy $P(r = i) \propto \min_{1 \le k' \le k} \|y_i - x_i \hat{B}_{k'}\|_2^2$
    $\hat{B}_k = \frac{x_r^\top y_r}{\|x_r\|_2^2}$
**end**
$t = 0$ **while** $\hat{u}_i$ *not converge* **do**
    **for** $k \leftarrow 1$ **to** $K$ **do**
        $X_k^{\mathcal{U}}$ is the matrix whose rows are all $x_i | \hat{u}_i = k$
        $Y_k^{\mathcal{U}}$ is the matrix whose rows are all $y_i | \hat{u}_i = k$
        We seem $\hat{B}_k$ as a $p * q$ dimensional matrix and
        $vec(\hat{B}_k) \sim N(vec((X_k^{u\top} X_k^u + \frac{T_t}{2\kappa} I_p)^{-1} X_k^{u\top} Y_k^u), \frac{T_t}{2}(I_q \otimes X_k^{u\top} X_k^u + \frac{T_t}{2\kappa} I_{p \times q})^{-1})$
    **end**
    **for** $j \leftarrow 1$ **to** $n$ **do**
        $p(\hat{u}_j = k) \propto \exp(-\frac{\|y_j - x_j \hat{B}_k\|_2^2}{T_t})$
    **end**
    $t = t + 1$
    $T_t = T_0 \cdot (\log t)^{-\alpha}$, where $0 < \alpha < 1$
**end**

---

## 3 MAIN RESULTS

### 3.1 NOTATIONS AND ASSUMPTIONS

This section presents the main theoretical results for our algorithm. Our framework assumes that the model described in (1) is correct and the true number of classes $K$ is given. The core of our analysis is the WCSS function $J$ and the residual sum of squares function $m$. Consequently, the global minimizer of $J$ shares the same parameter estimate $\hat{\theta}$ as the empirical mean $\frac{1}{n} \sum_{i=1}^n m(x_i, y_i, \hat{\theta})$. To analyze the properties of the global minimum of $J$, we must therefore examine both the empirical objective $\frac{1}{n} \sum_{i=1}^n m(x_i, y_i, \theta)$ and its population counterpart $\mathbb{E}_{(X,Y)}[m(X, Y, \theta)]$. On the other hand, for the model equation 1 to yield statistically significant conclusions, nondegeneracy is essential. We thus formalize the following assumptions before the theoretical analysis.

**Assumption 1** (Uniqueness of optimal solution up to permutation symmetry). *The global minimum $\hat{\theta} = (\hat{B}_1, \hat{B}_2, \cdots, \hat{B}_K)$ of the function $\frac{1}{n} \sum_{i=1}^n m(x_i, y_i, \theta)$ and the global minimum $\theta^* = (B_1^*, B_2^*, \cdots, B_K^*)$ of its expectation $\mathbb{E}_{(X,Y)}[m(X, Y, \theta)]$ are unique up to permutations. That is,*

*if $\hat{\theta}$ is a global minimum of $\frac{1}{n}\sum_{i=1}^{n} m(x_i, y_i, \theta)$, then any permutation $(\hat{B}_{\pi(1)}, \hat{B}_{\pi(2)}, \cdots, \hat{B}_{\pi(K)})$, for $\pi \in S_K$ (the symmetric group on the $K$ elements) is also a global minimum, and all global minima are permutations of each other. Similarly, this property holds for the population minimizer $\theta^* = (B_1^*, B_2^*, \cdots, B_K^*)$*

**Assumption 2** (Model correctness). *The distribution of $X, Y$ fits the model (equation 1), with $p_k \geq c > 0$ for any $1 \leq k \leq K$.*

**Assumption 3** (Model non-degeneracy). *The covariance matrix $\Sigma$ of the variable $X$ is non-degenerate, which means that the minimum eigenvalue of $\Sigma$ is strictly greater than zero.*

Our subsequent analysis relies on the three assumptions mentioned previously. We define $\mathcal{U}^* = \{u_1^*, \cdots, u_n^*\}$, where each assignment $u_i^*$ is the index of the true parameter that minimizes the squared error, that is, $u_i^* = \arg\min_{1 \leq k \leq K} \|y_i - x_i B_k^*\|_2^2$. Throughout, we use $\|\cdot\|_F$ to denote the Frobenius norm and $\|\cdot\|_{min}$ for the minimum eigenvalue of a symmetric positive definite matrix.

### 3.2 THEOREMS ABOUT ESTIMATED QUALITY AND CLASSIFICATION ACCURACY

This subsection establishes an upper bound on the estimation error between the global minimum of $\mathbb{E}_{(X,Y)}[m(X, Y, \theta)], \theta^* = (B_1^*, B_2^*, \cdots, B_K^*)$ and the true parameters $\theta = (B_{1,0}, B_{2,0}, \cdots, B_{K,0})$, under the assumption that the number of clusters $K$ is known. By analyzing the structural properties of the objective function $m(X, Y, \theta)$, we derive a high-probability bound for the parameter estimation error. This result further implies a limit on classification accuracy, defined as $\frac{1}{n}\sum_{i=1}^{n} I(u_i^* = \pi(u_{i,0}))$, where $\pi$ denotes the optimal permutation that aligns the estimated cluster parameters with their true counterparts.

The quality of the parameter estimates is fundamental to the overall classification performance. Therefore, we first conduct a thorough analysis of $m(X, Y, \theta)$ to control the estimation error. The main theoretical contribution is presented in Lemma 3.1, which provides an upper bound on the gap of the regression matrices of $\theta^*$ and $\theta$ in the large sample setting. This bound explicitly characterizes how the accuracy of the estimate depends on the dimensions of the problem $(p, q)$ and the spectral properties of the covariance matrices $\sigma$ and $\Sigma$, providing information on the factors driving the accuracy of the estimate.

**Lemma 3.1.** *Under Assumptions 1, 2 and 3, we denote the estimator $\theta^* = B_1^*, B_2^*, \cdots, B_K^*$ minimize the*

$$\mathbb{E}_{(X,Y)}[m(X, Y, \theta)].$$

*Then we have for any $1 \leq k \leq K$, there exist a $1 \leq \pi(k) \leq K$ satisfy:*

$$\|B_{k,0} - B_{\pi(k)}^*\|_F \leq C \frac{\sigma}{\sqrt{\|\Sigma\|_{min}}},$$

*where*

$$C = K\sqrt{3e}\{(K-1)\sqrt{\frac{2}{\pi}} + \sqrt{\frac{1}{c}\frac{2}{\pi}(K-1)^2 + \frac{1}{c}2(K-1)\sqrt{\frac{2}{\pi}}}\},$$

*which only related to the $K$ and c.*

Lemma 3.1 establishes an asymptotic upper bound on the Frobenius norm error $\|B_{k,0} - B_{\pi(k)}^*\|_F$ for each true cluster parameter $B_{k,0}$ and the minimum point $B_{\pi(k)}^*$ of function $m$. This bound characterizes the behavior of the estimator $\hat{\theta}$ in large samples, with its magnitude governed by the noise level $\sigma$, the minimum singular value of the covariance matrix $\|\Sigma\|_{min}$, and a constant $C$ that depends solely on $K$ and $c$. Notice that for $i \neq j$, we cannot prove $\pi(i) \neq \pi(j)$ without further conditions.

The precision of these parameter estimates is fundamental to the classification accuracy, defined as $\frac{1}{n}\sum_{i=1}^{n} \mathbb{I}(u_i^* = \pi(u_{i,0}))$. Intuitively, accurate classification requires that the maximum estimation error, $\max_{1 \leq k \leq K} \|B_{k,0} - B_{\pi(k)}^*\|_F$, is small relative to the minimum separation between two distinct true parameters, $D = \min_{i \leq j} \|B_{i,0} - B_{j,0}\|_F$. When this condition is satisfied, that is, when the estimated parameters are close to their true values and the clusters are well separated, the probability of misclassification decays rapidly. The bound in Lemma 3.1 provides a direct pathway to formalize this intuition and derive a subsequent bound on the classification error rate.

In particular, if the distance between classes $D$ between the different regression matrices is greater than twice the maximum coefficient estimation error, $\max_{1 \leq k \leq K} \|B_{k,0} - \hat{B}_{\pi(k)}\|_F$, it can be proven that $\pi(i) \neq \pi(j)$ for any $i \neq j$. This implies that $\pi \in S_K$ is a true permutation, which prevents a single estimated matrix from being matched to multiple true matrices. Under slightly stronger conditions, Theorem 3.2 establishes that, in the asymptotic sense of large samples, the probability of missclassification decays at a rate $\mathcal{O}_p\left(D^{-1}(\log D)^{1/2}\right)$. This shows that in the sense of large samples, when the degree of separation between categories $D$ tends to infinity, the probability that each sample is correctly classified tends to 1.

**Theorem 3.2.** *Let* $D' = \frac{\sqrt{\|\Sigma\|_{min}}}{\sigma}D$ *and* $C' = \sqrt{\frac{\|\Sigma\|_2}{\|\Sigma\|_{min}}}C$. *Under Assumptions 1, 2 and 3, if the inequality condition* $D' > 2C' + 2\sqrt{q}$ *holds, then, for any sample* $y_j = B_{k,0}x_j + \epsilon_j$ *from sub-distribution* $Y = B_{k,0}X + \epsilon$, *the probability that this sample is correctly clustered could be bounded by:*

$$P(u_{j,0} \neq \pi(k)) \leq (K-1)\{\frac{C' + 2\sqrt{q}}{D' - C'}(1 + 2\log(\frac{D' - C'}{C' + 2\sqrt{q}}))^{\frac{1}{2}}$$

$$+(\frac{C' + 2\sqrt{q}}{D' - C'}(1 + 2\log(\frac{D' - C'}{C' + 2\sqrt{q}}))^{\frac{1}{2}})^q + e^{\frac{1}{2}}(\frac{C' + 2\sqrt{q}}{D' - C'})(1 + 2\log(\frac{D' - C'}{C' + 2\sqrt{q}}))^{\frac{1}{2}}\}$$

Theorem 3.2 gives an upper bound on the probability that each sample is incorrectly clustered. This upper bound tends to 0 at a rate of $\mathcal{O}_p\left(D^{-1}(\log D)^{1/2}\right)$ when the degree of separation $D$ of the regression matrices of different sub-models tends to infinity. Through the precise guarantee on the accuracy of sample classification provided by Theorem 3.2, when $D$ is large, we can obtain a more precise upper bound guarantee on the error of estimating the true parameter $\theta$ using the global minimum $\theta^*$ than Lemma 3.1. In fact, in Theorem 3.3, we proved that when $D$ tends to infinity, the estimation error $\max_{1 \leq k \leq K} \|B_{k,0} - B_k^*\|_F$ decreases at the rate $\mathcal{O}_p\left(D^{-\frac{1}{2}}(\log D)^{1/4}\right)$

**Theorem 3.3.** *Denoting* $D' = \frac{\sqrt{\|\Sigma\|_{min}}}{\sigma}D$ *and* $C' = \sqrt{\frac{\|\Sigma\|_2}{\|\Sigma\|_{min}}}C$. *Under Assumptions 1, 2 and 3, if the inequality conditions* $D' > 2C' + 2$ *and* $(K-1)(1 + e^{\frac{1}{2}})\frac{C' + 2}{D' - C'}(1 + 2\log(\frac{D' - C'}{C' + 2}))^{\frac{1}{2}} \leq 1$ *holds and estimator* $\theta^* = (B_1^*, B_2^*, \cdots, B_K^*)$ *minimize the* $\mathbb{E}_{X,Y}m(X, Y, \theta)$, *then there is a constant* $C_D$, *for any* $1 \leq k \leq K$, *there exist a* $\pi(k)$ *satisfies*

$$\|B_{k,0} - B_{\pi(k)}^*\|_F \leq C_D \frac{\sigma}{\sqrt{\|\Sigma\|_{min}}}$$

*for any* $1 \leq k \leq K$, *where*

$$C_D = \frac{\sqrt{3e}}{1 - P(K-1)}\{(K-1)\sqrt{\frac{2}{\pi}}e^{-\frac{1}{2}}\frac{C' + 2}{D' - C'}$$

$$+\sqrt{\frac{1}{c}\frac{2}{e\pi}(K-1)^2(\frac{C' + 2}{D' - C'})^2 + \frac{1}{c}2(K-1)\sqrt{\frac{2}{\pi}}\frac{e^{-\frac{1}{2}}}{1 + e^{-\frac{1}{2}}}P + \frac{1}{c}\frac{P(K-1)}{1 - P(K-1)}(\frac{2}{\pi}(K-1)^2 + 2\sqrt{\frac{2}{\pi}}(K-1))}\}$$

*with*

$$P = (1 + e^{\frac{1}{2}})\frac{C' + 2}{D' - C'}(1 + 2\log(\frac{D' - C'}{C' + 2}))^{\frac{1}{2}}$$

*and* $C' = C\sqrt{\frac{\|\Sigma\|_2}{\|\Sigma\|_{min}}} = \sqrt{\frac{\|\Sigma\|_2}{\|\Sigma\|_{min}}}K\sqrt{3e}\{(K-1)\sqrt{\frac{2}{\pi}} + \sqrt{\frac{1}{c}\frac{2}{\pi}(K-1)^2 + \frac{1}{c}2(K-1)\sqrt{\frac{2}{\pi}}}\}$.

In summary, our analysis provides rigorous theoretical guarantees for the proposed framework by analyzing its properties at both the population and finite-sample levels. Our primary theoretical contribution is to establish that the population minimizer of our objective function, $\theta^*$, which represents the asymptotic properties of $\hat{\theta}$, is a consistent proxy for the true parameter $\theta_0$. We show that the asymptotic bias between $\theta^*$ and $\theta_0$ is primarily governed by the separation distance $D$, the bias, and the rate of misclassification decreasing as $D$ increases. This relationship is further influenced by the signal-to-noise ratio, where the bias is amplified by higher noise levels ($\sigma$) but reduced by a stronger signal structure (characterized by the spectral properties of $\Sigma$). For completeness, we provide a detailed analysis of the finite-sample error between $\hat{\theta}$ and $\theta_0$ in Appendix B. Collectively, these results guarantee the reliability of our method by showing that its theoretical target is provably close to the ground truth.

### 3.3 Theorems about algorithm convergence

In Subsection 3.2, we analyze the global minimum point $\theta^* = (B_1^*, B_2^*, \ldots, B_K^*)$ and investigate the statistical properties of this estimator to recover the true regression function and the true category labels. However, both the point-wise objective function $m(X, Y, \theta) = \min_{1 \leq k \leq K} \|Y - XB_k\|_2^2$ and its expectation are nonconvex. As a result, conventional K-means algorithms are generally unable to guarantee convergence to the global minimum. This section establishes that the proposed **Gibbs sampling with simulated annealing K-means clustering Algorithm** (Algorithm 1) converges provably to the global minimum of $\sum_{i=1}^n m(x_i, y_i, \theta)$ at a rate slightly slower than a power function. These convergence results underline the algorithmic advantage of incorporating stochastic sampling and annealing mechanisms to overcome the limitations of classical non-convex optimization in clustering contexts.

**Theorem 3.4.** *We denote $\hat{\mathcal{U}}^{(t)} = (u_1^{(t)}, \cdots, u_n^{(t)})$ and $\hat{\theta} = (\hat{B}_1^{(t)}, \cdots, \hat{B}_n^{(t)})$ as the estimation result of $t$-th iteration of Algorithm 1. If $\hat{\theta}$ is the global minimum of the function $\sum_{i=1}^n m(x_i, y_i, \theta)$, $\hat{\mathcal{U}} = (\hat{u}_1, \cdots, \hat{u}_n)$ is the category estimate generated by $\hat{\theta}$. If Assumption 1 holds and $T_{(1)} \leq T_{(2)}$ satisfies $T_{(1)}(\log t)^\alpha \leq T_t \leq T_{(2)}(\log t)^\alpha$ for $0 < \alpha < 1$, then there is a permutation $\pi$ such that for any $\delta > 0$,*

$$P(\hat{u}_i^{(t)} = \pi(\hat{u}_i)) \geq 1 - C^* \exp(-C^{**}(\log t)^\alpha),$$

$$P(\|\hat{B}_k^{(t)} - \hat{B}_{\pi(k)}\|_F < \delta) \geq 1 - C_\delta^* \exp(-C_\delta^{**}(\log t)^\alpha),$$

*where $C^*, C^{**} > 0$ are not related to $t$ or $\delta$, and $C_\delta^*, C_\delta^{**} > 0$ are not related to $t$.*

Theorem 3.4 establishes that the probability of convergence of our **Gibbs sampling with simulated annealing K-means clustering Algorithm** (Algorithm 1) to a neighborhood of the global minimum of the WCSS function $J(\theta, \mathcal{U})$ increases to 1 with the number of iterations. This result indicates that the algorithm converges with high probability and at a rapid rate to the WCSS minimum while remaining robust to initial conditions. Building on the theoretical framework developed in Section 3.2, these convergence guarantees imply that the algorithm produces estimates consistent with the true regression function and produces highly accurate predictions as well as classifications with high probabilities. In particular, these assurances hold for multivariate linear regression problems without reliance on overly restrictive assumptions.

## 4 Simulation studies

### 4.1 Simulation setup

This section presents a comprehensive empirical evaluation of the proposed **Gibbs sampling with simulated annealing K-means clustering Algorithm** (GIBBS-SA K-MEANS, or GSAKM) for multivariate linear regression, as formalized in Algorithm 1. Upon completing the iterative optimization procedure described in Algorithm 1, we lower the temperature $T_t$ to 0 for the final polishing. To mitigate convergence to local optima and improve the quality of the solution, we performed 10 independent optimization trials with random initializations under all experimental conditions. Throughout our experiments, the annealing parameter $\alpha$ is maintained at 0.99, a value empirically calibrated to strike a balance between exploration and exploitation during the optimization process.

To improve convergence probability and reduce the number of iterations required, we adopt a temperature scheduling scheme defined by $T_t = T(\log(t_0 + t) - t_1)^{-\alpha}$, where $t_0, t_1$ are parameters introduced to prevent an excessively rapid decrease in temperature during initial iterations. In particular, $T$ is not kept constant, but is instead dynamically scaled in proportion to the minimum value of $\sum_{i=1}^n m(x_i, y_i, \theta)$ observed in all iterations. Since $\sum_{i=1}^n m(x_i, y_i, \theta)$ has a global minimum, the decay rate of our temperature $T_t$ remains consistent with the conditions specified in Theorem 3.4. Specifically in our simulation studies, denoting $\hat{\theta}^{(s)}$ as the estimate parameter of $t$-th iteration, we set $T = \frac{K}{np - Kpq} \min_{1 \leq s \leq t} \sum_{i=1}^n m(x_i, y_i, \hat{\theta}^{(s)})$, $\kappa = 0.01$, $t_0 = 2\exp(4)$ and $t_1 = 3 + \log(2)$

To establish comparative baselines, we evaluate our proposed methodology (Algorithm 1) against three established approaches: standard Expectation-Maximization (SEM), a variant of EM that assumes a known error variance $\sigma^2$ (SEMK), and standard K-Means clustering (SKM). Our simulation framework generates data from the Gaussian mixture model specified in Equation equation 1 with

a fixed sample size of $n = 500$. The covariate vectors $x_i \in \mathbb{R}^p$ are sampled from $\mathcal{N}(0, \Sigma)$, where $\Sigma$ has an autoregressive covariance structure with $\Sigma_{ij} = 0.3^{|i-j|}$. The error terms are extracted independently from $\epsilon_i \sim \mathcal{N}(0, I_q)$ (with $\sigma = 1$), and the response variables $y_i$ are subsequently derived from the mixture model.

The experimental design systematically evaluates performance across multiple dimensions of problem complexity. We test both three-cluster ($K = K = 3$) systems with predictor dimensions of $p \in \{50, 70\}$ and four-cluster ($K = K = 4$) systems with $p \in \{35, 50\}$. For each of these $(K, p)$ pairs, we further vary the response dimensionality to include the $q \in \{2, 3\}$ variables. For each resulting combination, we then evaluate the cases with regression dimensions of $D \in \{20, 40\}$. This complete factorial design yields a total of $2 \times 2 \times 2 \times 2 = 16$ unique experimental conditions. All four algorithms —-SEM, SEMK, SKM, and GSAKM —- undergo a rigorous evaluation under each parameter configuration, enabling a comprehensive assessment of their relative performance advantages across these varying complexities.

## 4.2 SIMULATION RESULTS

To evaluate prediction methods in multivariate linear regression with mixture models, we employ two metrics: estimation error and classification accuracy. The estimation error is defined as $\min_{\pi \in \mathcal{S}_K} \max_{1 \le k \le K} \|\hat{B}_{\pi(k)} - B_{k,0}\|_F$, where $\mathcal{S}_K$ denotes the symmetric group of all permutations of $\{1, 2, \cdots, K\}$. This permutation minimization accounts for label switching, ensuring invariance to class relabeling. Notice that this definition remains valid regardless of whether the conditions in Theorems in Subsection 3.2 hold, eliminating the lower bound assumptions about $D$ in numerical experiments. The classification accuracy is $\frac{1}{n} \sum_{i=1}^n \mathbb{I}(\hat{u}_i = \pi(u_{i,0}))$. The predicted label $\hat{u}_i$ is determined by $\hat{u}_i = \arg\min_{1 \le k \le K} \|y_i - x_i \hat{B}_k\|_2$.

Beyond estimating the regression parameter $\hat{\theta}$ and assigning group memberships $\{\hat{u}_i\}_{1 \le i \le n}$ from the training data to compute prediction and classification errors, we perform additional validation using an independently generated testing set. This test dataset, simulated from the same model with an identical sample size ($n = 500$), allows for the calculation of the out-of-sample classification error. For both training and testing datasets, we further evaluated performance using WCSS, which is denoted by $J(\hat{\theta}, \hat{\mathcal{U}}) = \sum_{i=1}^n m(x_i, y_i, \hat{\theta})$.

This paper proposes a novel algorithm, combining Gibbs sampling with simulated annealing K-means, for the estimation of the mixture of multivariate linear regression models. We provide a comprehensive theoretical analysis that establishes that, under mild assumptions and sufficient separation ($D$) between the true regression matrices, the global minimizer of the objective function is a consistent estimator. Specifically, we prove that both the parameter estimation error and the misclassification rate converge to zero as $D$ increases. Algorithmically, we show that our method converges to this global minimum with high probability under a slow logarithmic cooling schedule with an exponent $\alpha < 1$.

The efficacy of our approach and its theoretical guarantees are validated through extensive experiments on both synthetic and real-world datasets. Although our theory is presented for standard errors, the framework is flexible enough to accommodate other distributions. Promising directions for future work include extending this analysis to more general parametric families, such as generalized linear models (GLMs), or to models based on soft component assignments.

## 5 DISCUSSION

This paper studies a mixed multivariate linear regression model using a Gibbs sampling-enhanced simulated annealing K-means clustering algorithm. We establish that, under mild assumptions, in both asymptotic and non-asymptotic (finite-sample) regimes, the global minimizer of the K-means objective accurately recovers the true regression matrix in finite samples and assigns observations to their true categories with high probability. Moreover, as the separation $D$ between different regression matrices increases, the parameter estimation error converges asymptotically to zero, and the misclassification rate decays asymptotically to zero. Algorithmically, we prove that under a logarithmic cooling schedule with exponent $\alpha < 1$, the probability of converging to the global minimum behaves as $(\log t)^{-\alpha}$. Although the theory assumes standard errors, the framework is

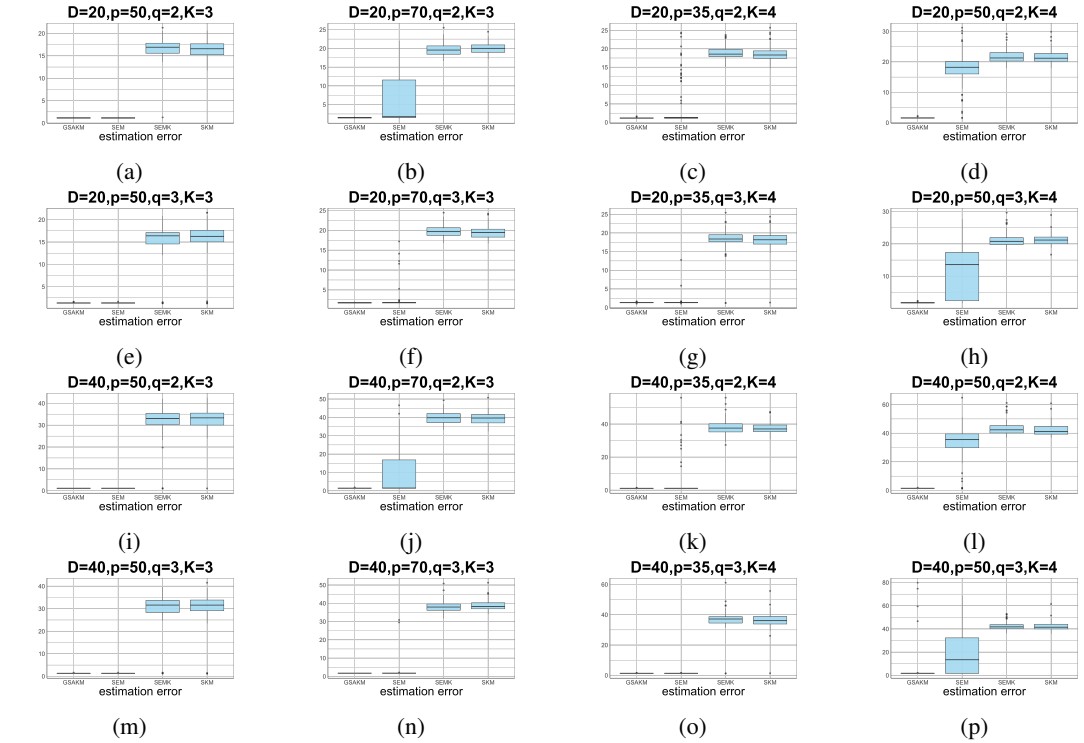

Figure 1: The box-plot of estimation errors of four different estimation methods under 16 parameter conditions.

flexible and can accommodate other distributions. Empirical results, based on both synthetic and real data, support our theoretical claims. Future work could extend our analysis to more general parametric families, such as Generalized Linear Models (GLMs), or develop estimation methods for models where observations arise from soft assignments or linear combinations of the underlying components.

ETHICS STATEMENT

This work adheres to the ICLR Code of Ethics. As a foundational and theoretical study validated on synthetic data, it presents no direct ethical risks involving human subjects or sensitive information. However, we encourage careful consideration of fairness and bias in any real-world application of this general-purpose algorithm.

REPRODUCIBILITY STATEMENT

We are committed to ensuring the reproducibility of the research presented in this paper. All code, simulation scripts, and instructions required to replicate the experiments, figures, and tables are provided in the supplementary material.

**Code**  The implementation of our experiments was carried out in R (version 4.4.3). To guarantee a fully reproducible software environment, we have utilized the `renv` package. The exact versions of all R packages are captured in the `renv.lock` file. Detailed setup instructions are available in the `README.md` file included in our submission. The main simulation logic can be found in `simulate_study_program.R`, with five figures generation scripts located in the leading directory.

**Data**  All datasets analyzed in this work were generated by simulation. The code for this data generation process is an integral part of the main simulation script (`simulate_study_program.R`), enabling the complete end-to-end replication of our results, from data creation to final analysis.

LARGE LANGUAGE MODEL ASSISTANCE

During the preparation of our code and supplementary materials for submission, we utilized Google's Gemini and Deepseek. Its assistance was specifically sought for debugging R code, improving the language and clarity of the main text, and refining technical descriptions. The authors assume full and final responsibility for all content presented in this paper and its supplementary materials.

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

# A THE PROOF OF THE THEOREM GIVEN IN THE PAPER

## A.1 PROOF OF LEMMA 3.1

*Proof.* For any fix $X$,we define $Y_k^* = XB_k^*$ for $1 \leq k \leq K$ and $Y_k = XB_{k,0}$ for $1 \leq k \leq K$ then $\mathbb{E}_{(X,Y)}[m(X,Y,\theta^*)] = \mathbb{E}_X(\mathbb{E}_Y \min_{1 \leq k' \leq K} \|Y - Y_{k'}^*\|_2^2)$. Here $Y$ follows a mixture Gaussian model, for probability $p_k$, $Y \sim N(Y_k, \sigma^2 I_q)$. So we have $\mathbb{E}_X(\mathbb{E}_Y \min_{1 \leq k' \leq K} \|Y - Y_{k'}^*\|_2^2) = \mathbb{E}_X(\sum_{k=1}^n p_k \mathbb{E}_{Y \sim N(Y_k, \sigma^2 I_q)} \min_{1 \leq k' \leq K} \|Y - Y_{k'}^*\|_2^2)$. Because $X$ is not related to the mixing ratio. We have $\mathbb{E}_X(\mathbb{E}_Y \min_{1 \leq k' \leq K} \|Y - Y_{k'}^*\|_2^2) = \sum_{k=1}^n p_k \mathbb{E}_X(\mathbb{E}_{Y \sim N(Y_k, \sigma^2 I_q)} \min_{1 \leq k' \leq K} \|Y - Y_{k'}^*\|_2^2)$

We let $t_k = min\{\|Y_{k'}^* - Y_k\|_2, 1 \leq k' \leq K\}$. It may be worthwhile to let $t_k = \|Y_1^* - Y_k\|_2$. Then if $Y \sim N(Y_k, \sigma^2 I_q)$, we have:

$$\mathbb{E}_{Y \sim N(Y_k, \sigma^2 I_q)} \min_{1 \leq k' \leq K} \|Y - Y_{k'}^*\|_2^2$$

$$= \mathbb{E}_{Y \sim N(Y_k, \sigma^2 I_q)} \min_{2 \leq k' \leq K}(min(\|Y - Y_1^*\|_2^2, \|Y - Y_{k'}^*\|_2^2)$$

$$= \mathbb{E}_{Y \sim N(Y_k, \sigma^2 I_q)} \min_{2 \leq k' \leq K}(\|Y - Y_1^*\|_2^2 - \max\{0, \|Y - Y_1^*\|_2^2 - \|Y - Y_{k'}^*\|_2^2\})$$

$$\geq \mathbb{E}_{Y \sim N(Y_k, \sigma^2 I_q)}(\|Y - Y_1^*\|_2^2 - \sum_{k'=2}^{K} \max\{0, \|Y - Y_1^*\|_2^2 - \|Y - Y_{k'}^*\|_2^2\})$$

Notice that $\|Y - Y_1^*\|_2^2 - \|Y - Y_{k'}^*\|_2^2 = 2 < Y_{k'}^* - Y_1^*, Y - \frac{1}{2}(Y_1^* + Y_{k'}^*) >$. We let $t_k' = \|Y_{k'} - Y_k\|_2$. Then $\mathbb{E}_{Y \sim N(Y_k, \sigma^2 I_q)} \max\{0, 2 < Y_{k'}^* - Y_1^*, Y - \frac{1}{2}(Y_1^* + Y_{k'}^*) >\} \leq \mathbb{E}_{y \sim N(0,\sigma)} 2(t_k + t_k') \max\{0, y - \frac{t_k' - t_k}{2}\}$

By calculation, we have

$$\mathbb{E}_{y \sim N(0,\sigma)} \max\{0, y - \frac{t_k' - t_k}{2}\} =$$

$$\int_{t=\frac{t_k' - t_k}{2}}^{+\infty} \frac{1}{\sqrt{2\pi}\sigma}(t - \frac{t_k' - t_k}{2}) e^{-\frac{t^2}{2\sigma^2}} dt$$

$$\leq \int_{t=\frac{t_k' - t_k}{2}}^{+\infty} \frac{1}{\sqrt{2\pi}\sigma} t e^{-\frac{t^2}{2\sigma^2}} dt$$

$$= \frac{\sigma}{\sqrt{2\pi}} e^{-\frac{a^2}{2\sigma^2}}$$

Where $a = \frac{t_k' - t_k}{2}$ Because of $t_k' \geq t_k$, we have $a \geq 0$, and $\mathbb{E}_{y \sim N(0,\sigma)} 2(t_k + t_k') \max\{0, y - \frac{t_k' - t_k}{2}\} = 2(t_k + t_k') \frac{\sigma}{\sqrt{2\pi}} e^{\frac{-a^2}{2\sigma^2}} = 4\frac{\sigma}{\sqrt{2\pi}}(t_k + a) e^{-\frac{a^2}{2\sigma^2}}$

When $a = \frac{-t_k + \sqrt{t_k^2 + 4\sigma^2}}{2}$, the $4\frac{\sigma}{\sqrt{2\pi}}(t_k + a) e^{-\frac{a^2}{2\sigma^2}}$ take it's maximin, which equals to $2\frac{\sigma}{\sqrt{2\pi}}(t_k + \sqrt{t_k^2 + 4\sigma^2}) e^{-\frac{(-t_k + \sqrt{t_k^2 + 4\sigma^2})^2}{8\sigma^2}}$. It is easy to proof this maximum is smaller than $\frac{4\sigma}{\sqrt{2\pi}}(t_k + \sigma)$

In summary, we have:

$$\mathbb{E}_{Y \sim N(Y_k, \sigma^2 I_q)} \min_{1 \leq k' \leq K} \|Y - Y_{k'}^*\|_2^2$$

$$\geq \mathbb{E}_{Y \sim N(Y_k, \sigma^2 I_q)}(\|Y - Y_1^*\|_2^2 - \sum_{k'=2}^{K} \max\{0, \|Y - Y_1^*\|_2^2 - \|Y - Y_{k'}^*\|_2^2\})$$

$$\geq q\sigma^2 + t_k^2 - (K-1)\frac{4\sigma}{\sqrt{2\pi}} t_k - (K-1)\frac{4\sigma^2}{\sqrt{2\pi}}$$

So in mixed distribution, we have:

$$\sum_{k=1}^{n} p_k \mathbb{E}_{Y \sim N(Y_k, \sigma^2 I_q)} \min_{1 \leq k' \leq K} \|Y - Y_{k'}^*\|_2^2$$

$$\geq q\sigma^2 + \sum_{k=1}^{n} p_k (t_k^2 - (K-1)\frac{4\sigma}{\sqrt{2\pi}} t_k - (K-1)\frac{4\sigma^2}{\sqrt{2\pi}})$$

Now consider the variable of $X$, $t_k = min\{\|Y_{k'}^* - Y_k\|_2, 1 \leq k' \leq K\} = min\{\|XB_{k'}^* - XB_{k,0}\|_2, 1 \leq k' \leq K\}$ vary with $X$. So, it is easy to prove, based on the Jensen inequality, that:

$$q\sigma^2 + \sum_{k=1}^{n} p_k (t_k^2 - (K-1)\frac{4\sigma}{\sqrt{2\pi}} t_k - (K-1)\frac{4\sigma^2}{\sqrt{2\pi}})$$

$$\geq q\sigma^2 + \sum_{k=1}^{n} p_k (\mathbb{E}_X(t_k^2) - (K-1)\frac{4\sigma}{\sqrt{2\pi}} \sqrt{\mathbb{E}_X(t_k^2)} - (K-1)\frac{4\sigma^2}{\sqrt{2\pi}})$$

Now we give an upper bound of $\mathbb{E}_X(t_k^2)$ by rewrite $\|XB_{k'}^* - XB_{k,0}\|_2^2 = \sum_{i=1}^{p} a_i \xi_i^2$ where $a_i$ is the $i$th eigenvalue of matrix $(B_{k'}^* - B_{k,0})^\top \Sigma (B_{k'}^* - B_{k,0})$ for $1 \leq i \leq p, \xi_1, \xi_2, \cdots, \xi_p$ are independent standard normal distributed random variables. For any $\lambda > 0$ and $\mu > 0$ we have:

$$P(\sum_{i=1}^{p} a_i \xi_i^2 \leq \lambda) = P(e^{-\mu \sum_{i=1}^{p} a_i \xi_i^2} \geq e^{-\mu \lambda})$$

$$\leq e^{\mu \lambda} E e^{-\mu \sum_{i=1}^{p} a_i \xi_i^2}$$

$$\leq e^{\mu \lambda} \prod_{i=1}^{p} E e^{-\mu a_i \xi_i^2} = e^{\mu \lambda} \prod_{i=1}^{p} (1 + 2\mu a_i)^{-\frac{1}{2}} \leq e^{\mu \lambda} (1 + 2\mu \sum_{i=1}^{p} a_i)^{-\frac{1}{2}}$$

We take $\mu = \frac{1}{2}(\frac{1}{\lambda} - \frac{1}{\sum_{i=1}^{p} a_i})$ when $\lambda < \sum_{i=1}^{p} a_i$, we have $P(\|XB_{k'}^* - XB_{k,0}\|_2^2 \leq \lambda) \leq \sqrt{\frac{\lambda}{\sum_{i=1}^{p} a_i}} e^{\frac{1}{2}(1 - \frac{\lambda}{\sum_{i=1}^{p} a_i})} \leq \sqrt{\frac{e\lambda}{\sum_{i=1}^{p} a_i}}$

Notice that If $\|B_{k'}^* - B_{k,0}\|_F = F_{k',k}$, we have $\sum_{i=1}^{p} a_i = tr((B_{k'}^* - B_{k,0})^\top \Sigma (B_{k'}^* - B_{k,0})) = tr(\Sigma(B_{k'}^* - B_{k,0})(B_{k'}^* - B_{k,0})^\top) \leq \|\Sigma\|_{min} F_{k',k}^2$

If $\pi(k)$ let $F_{\pi(k),k}$ is the minimum of $F_{k',k}$ when $1 \leq k' \leq K$, we have $P(t_k^2 \leq \lambda) = P(\min_{1 \leq k' \leq K} \|XB_{k'}^* - XB_{k,0}\|_2^2 \leq \lambda) \leq K\sqrt{\frac{e\lambda}{\|\Sigma\|_{min} F_{\pi(k),k}^2}}$, so we have $E(t_k^2) \geq \int_{\lambda=0}^{\frac{\|\Sigma\|_{min} F_{\pi(k),k}^2}{eK^2}} (1 - K\sqrt{\frac{e\lambda}{\|\Sigma\|_{min} F_{\pi(k),k}^2}}) d\lambda = \frac{\|\Sigma\|_{min} F_{\pi(k),k}^2}{3eK^2}$

So, if for any $1 \leq k \leq K$,

$$F_{\pi(k),k} > \frac{\sigma K \sqrt{3e}}{\sqrt{\|\Sigma\|_{min}}} \{(K-1)\sqrt{\frac{2}{\pi}} + \sqrt{\frac{1}{c}\frac{2}{\pi}(K-1)^2 + \frac{1}{c}2(K-1)\sqrt{\frac{2}{\pi}}}\}.$$

Then we have

$$\mathbb{E}_{Y \sim N(Y_k, \sigma^2 I_q)} \min_{1 \leq k' \leq K} \|Y - Y_{k'}^*\|_2^2$$

$$> \sigma^2(q + \frac{2}{\pi}\frac{1-c}{c}(K-1)^2 + 2\sqrt{\frac{2}{\pi}}\frac{1-c}{c}(K-1))$$

so

$$\sum_{k=1}^{K} p_k \mathbb{E}_{Y \sim N(Y_k, \sigma^2 I_q)} \min_{1 \leq k' \leq K} \|Y - Y_{k'}^*\|_2^2$$

$$> \sum_{k=1}^{K} \sigma^2(q + p_k \frac{2}{\pi}\frac{1-c}{c}(K-1)^2 + 2p_k \sqrt{\frac{2}{\pi}}\frac{1-c}{c}(K-1)$$

$$- (1-p_k)\frac{2}{\pi}(K-1)^2 - (1-p_k)\frac{4}{\sqrt{2\pi}}(K-1)) \geq q\sigma^2$$

It is easy to prove when $B_{k'}^* = B_{k,0}$ for $k' \leq K$, $\mathbb{E}_{Y \sim N(Y_k, \sigma^2 I_q)} \min_{1 \leq k' \leq K} \|Y - Y_{k'}^*\|_2^2 \leq q\sigma^2$, so if the $\theta^*$ minimize the $\mathbb{E}_{(X,Y)}[m(X, Y, \theta^*)]$, there exist a $\pi(k)$ satisfy: $\|B_{k,0} - B_{\pi(k)}^*\|_F \leq \frac{\sigma K \sqrt{3e}}{\sqrt{\|\Sigma\|_{min}}}\{(K-1)\sqrt{\frac{2}{\pi}} + \sqrt{\frac{1}{c}\frac{2}{\pi}(K-1)^2 + \frac{1}{c}2(K-1)\sqrt{\frac{2}{\pi}}}\}$ for any $k$ □

## A.2 PROOF OF THEOREM 3.2

*Proof.* Let's assume that under the best matching, $\pi(k) = k$ Using the conclusion of Lemma 3.1, we have $\|B_{k,0} - B_k^*\|_F < C\frac{\sigma}{\sqrt{\|\Sigma\|_{min}}} = C'\frac{\sigma}{\sqrt{\|\Sigma\|_2}}$. The conditions of $D$ tell us for any $i \neq k$, $\|B_{k,0} - B_i^*\|_F > (D' - C')\frac{\sigma}{\sqrt{\|\Sigma\|_{min}}}$.

If $D' > 2C'$, it is easy to see that for any $i \neq k$, $\|B_{k,0} - B_i^*\|_F > \|B_{k,0} - B_k^*\|_F$.

Then, if the sample $y_j = B_{k,0}x_j + \epsilon_j$ is from the sub-distribution $Y = B_{k,0}X + \epsilon$, then if $u_j^* = i$, it means $\|y_j - B_i^* x_j\|_2 < \|y_j - B_k^* x_j\|_2$, so $\|\epsilon_j + (B_{k,0} - B_i^*)x_j\|_2 < \|\epsilon_j + (B_{k,0} - B_k^*)x_j\|_2$.

On the other hand, because of $\|B_{k,0} - B_i^*\|_F > (D' - C')\frac{\sigma}{\sqrt{\|\Sigma\|_{min}}}$ and $\|B_{k,0} - B_k^*\|_F < C'\frac{\sigma}{\sqrt{\|\Sigma\|_2}}$, using the tail bound of chi-square distribution we have use in the proof of theorem 3.1, we have:

$$\mathbb{P}\left(\|(B_{k,0} - B_i^*)x_j\|_2^2 < (D' - C')^2\sigma^2 - \lambda\right) \le e^{\frac{1}{2}\frac{\lambda}{(D'-C')^2\sigma^2}}(1 - \frac{\lambda}{(D' - C')^2\sigma^2})^{\frac{1}{2}}$$

and

$$\mathbb{P}\left(\|(B_{k,0} - B_k^*)x_j\|_2^2 > C'^2\sigma^2 + \lambda\right) \le e^{-\frac{1}{2}\frac{\lambda}{C'^2\sigma^2}}(1 + \frac{\lambda}{C'^2\sigma^2})^{\frac{1}{2}}$$

satisfy for any $\lambda > 0$

And for error term $\epsilon_j$, beacuse $\epsilon_j \sim N(0, \sigma I_q)$, it is clear that $\frac{\|\epsilon_j\|_2^2}{\sigma^2} \sim \chi^2(q)$, so the tail bound of $\|\epsilon_j\|_2^2$ is:

$$\mathbb{P}\left(\|\epsilon_j\|_2^2 > q\sigma^2 + \lambda\right) \le (1 + \frac{\lambda}{q\sigma^2})^{\frac{q}{2}}e^{-\frac{\lambda}{2\sigma^2}}$$

Notice that, if $\|(B_{k,0} - B_i^*)x_j\|_2 - \|(B_{k,0} - B_k^*)x_j\|_2 \ge 2\|\epsilon_j\|_2$, then $\|y_j - B_i^* x_j\|_2 \ge \|y_j - B_k^* x_j\|_2$, which means $u_j^* \ne i$. So, we have for any $i \ne k$

$$\mathbb{P}(u_i^* = i) \le \mathbb{P}(\|(B_{k,0} - B_i^*)x_j\|_2 - \|(B_{k,0} - B_k^*)x_j\|_2 < 2\|\epsilon_j\|_2)$$

$$\le \mathbb{P}\left(\|(B_{k,0} - B_k^*)x_j\|_2^2 > C'^2\sigma^2 + 2C'^2\sigma^2\log(\frac{D' - C'}{C' + 2\sqrt{q}})\right)$$

$$+\mathbb{P}\left(\|\epsilon_j\|_2^2 > q\sigma^2 + 2q\sigma^2\log(\frac{D' - C'}{C' + 2\sqrt{q}})\right)$$

$$+\mathbb{P}\left(\|(B_{k,0} - B_i^*)x_j\|_2^2 < (C' + 2\sqrt{q})^2(1 + 2\log(\frac{D' - C'}{C' + 2\sqrt{q}}))\sigma^2\right)$$

$$\le \frac{C' + 2\sqrt{q}}{D' - C'}(1 + 2\log(\frac{D' - C'}{C' + 2\sqrt{q}}))^{\frac{1}{2}} + (\frac{C' + 2\sqrt{q}}{D' - C'}(1 + 2\log(\frac{D' - C'}{C' + 2\sqrt{q}}))^{\frac{1}{2}})^q$$

$$+e^{\frac{1}{2}}(\frac{C' + 2\sqrt{q}}{D' - C'})(1 + 2\log(\frac{D' - C'}{C' + 2\sqrt{q}}))^{\frac{1}{2}}$$

$$\square$$

## A.3 PROOF OF THEOREM 3.3

*Proof.* The condition we have is $\min_{1\le i<j\le k}\|B_{i,0} - B_{j,0}\|_F = D = D'\frac{\sigma}{\sqrt{\|\Sigma\|_{min}}}$, $\|B_{k,0} - B_k^*\|_F = \max_{1\le k\le K}\|B_{k,0} - B_k^*\|_F \le C\frac{\sigma}{\sqrt{\|\Sigma\|_{min}}} = C'\frac{\sigma}{\sqrt{\|\Sigma\|_2}}$ and $D' > 2C' + 2$.

Then, we use the definition in the proof of Lemma 3.1, denote $Y_k^* = XB_k^*$ and $Y_k = XB_{k,0}$ for $1 \le k \le K$, $t_k = \min_{1\le k'\le K}\|Y_k - Y_{k'}^*\|_2$ for ' $X$. With the condition we have, for any $k' \ne k$ we can proof:

$$\mathbb{P}\left(\|Y_{k'} - Y_k^*\|_2 - \|Y_k - Y_k^*\|_2 < 2\sigma(1 + 2\log(\frac{D' - C'}{C' + 2}))^{\frac{1}{2}}\right)$$

$$= \mathbb{P}\left(\|X(B_{k,0} - B_{k'}^*)\|_2 - \|X(B_{k,0} - B_k^*)\|_2 < 2\sigma(1 + 2\log(\frac{D' - C'}{C' + 2}))^{\frac{1}{2}}\right)$$

$$\le \mathbb{P}\left(\|X(B_{k,0} - B_k^*)\|_2^2 > C'^2\sigma^2(1 + 2\log(\frac{D' - C'}{C' + 2}))\right)$$

$$+\mathbb{P}\left(\|X(B_{k,0} - B_{k'}^*)\|_2^2 < (C' + 2)^2\sigma^2(1 + 2\log(\frac{D' - C'}{C' + 2}))\right)$$

$$\leq \frac{C'+2}{D'-C'}(1+2\log(\frac{D'-C'}{C'+2}))^{\frac{1}{2}} + e^{\frac{1}{2}}\frac{C'+2}{D'-C'}(1+2\log(\frac{D'-C'}{C'+2}))^{\frac{1}{2}}$$

$$= (1+e^{\frac{1}{2}})\frac{C'+2}{D'-C'}(1+2\log(\frac{D'-C'}{C'+2}))^{\frac{1}{2}}$$

So, the probability of $t_k = \|Y_k - Y_k^*\|_2$ could be bound, and for any $k' \neq k, t_k' = \|Y_k - Y_{k'}^*\|_2, a = \frac{t_k'-t_k}{2}$ we have:

$$\mathbb{P}\left(t_k = \|Y_k - Y_k^*\|_2, \text{ and } a > \sigma(1+2\log(\frac{D'-C'}{C'+2}))^{\frac{1}{2}} \text{ for any } k' \neq k\right)$$

$$\geq 1 - (K-1)(1+e^{\frac{1}{2}})\frac{C'+2}{D'-C'}(1+2\log(\frac{D'-C'}{C'+2}))^{\frac{1}{2}}$$

We denote $P = (1+e^{\frac{1}{2}})\frac{C'+2}{D'-C'}(1+2\log(\frac{D'-C'}{C'+2}))^{\frac{1}{2}}$ and reuse the conclusion obtained in the proof of Lemma 3.1. The global minimum of $\mathbb{E}_{X,Y}(X,Y,\theta)$ satisfy:

$$\mathbb{E}_{X,Y}(X,Y,\theta^*)$$

$$= \mathbb{E}_X \sum_{k=1}^{K} p_k \mathbb{E}_{Y \sim N(Y_k,\sigma^2 I_q)} \min_{1 \leq k' \leq K} \|Y - Y_{k'}^*\|_2^2$$

$$\geq \mathbb{E}_X \sum_{k=1}^{K} p_k \mathbb{E}_{Y \sim N(Y_k,\sigma^2 I_q)} (\|Y - Y_k^*\|_2^2 - \sum_{k' \neq k} \max\{0, \|Y - Y_k^*\|_2^2 - \|Y - Y_{k'}^*\|_2^2\})$$

$$\geq \mathbb{E}_X \sum_{k=1}^{K} p_k (q\sigma^2 + t_k^2 - 4(K-1)\frac{\sigma}{\sqrt{2\pi}}(t_k+a)e^{-\frac{a^2}{2\sigma^2}})$$

For any $k$, for at most probability $(K-1)P$, $t_k \neq \|Y_k - Y_k^*\|_2$ or $a \leq \sigma(1+2\log(\frac{D'-C'}{C'+2}))^{\frac{1}{2}}$, so $t_k^2 - 4(K-1)\frac{\sigma}{\sqrt{2\pi}}(t_k+a)e^{-\frac{a^2}{2\sigma^2}} \geq t_k^2 - 4\frac{\sigma}{\sqrt{2\pi}}(K-1)t_k - 4\frac{\sigma^2}{\sqrt{2\pi}}(K-1) \geq -\frac{2\sigma^2}{\pi}(K-1)^2 - \sigma^2\sqrt{\frac{2}{\pi}}(K-1)$. Otherwise, we have $t_k = \|Y_k - Y_k^*\|_2$ and $a > \sigma(1+2\log(\frac{D'-C'}{C'+2}))^{\frac{1}{2}}$, In this case, we have:

$$t_k^2 - 4(K-1)\frac{\sigma}{\sqrt{2\pi}}(t_k+a)e^{-\frac{a^2}{2\sigma^2}}$$

$$\geq t_k^2 - 4(K-1)\frac{\sigma}{\sqrt{2\pi}}e^{-\frac{1}{2}}\frac{C'+2}{D'-C'}t_k - 4(K-1)\frac{\sigma^2}{\sqrt{2\pi}}e^{-\frac{1}{2}}\frac{C'+2}{D'-C'}(1+2\log(\frac{D'-C'}{C'+2}))^{\frac{1}{2}}$$

$$= t_k^2 - 4(K-1)\frac{\sigma}{\sqrt{2\pi}}e^{-\frac{1}{2}}\frac{C'+2}{D'-C'}t_k - 4(K-1)\frac{\sigma^2}{\sqrt{2\pi}}\frac{e^{-\frac{1}{2}}}{1+e^{-\frac{1}{2}}}P$$

Using Jensen Inequality, under the condition of $t_k = \|Y_k - Y_k^*\|_2$ and $a > \sigma(1+2\log(\frac{D'-C'}{C'+2}))^{\frac{1}{2}}$, we have

$$\mathbb{E}_X[t_k^2 - 4(K-1)\frac{\sigma}{\sqrt{2\pi}}e^{-\frac{1}{2}}\frac{C'+2}{D'-C'}t_k - 4(K-1)\frac{\sigma}{\sqrt{2\pi}}\frac{e^{-\frac{1}{2}}}{1+e^{-\frac{1}{2}}}P]$$

$$\geq E(t_k^2) - 4(K-1)\frac{\sigma}{\sqrt{2\pi}}e^{-\frac{1}{2}}\frac{C'+2}{D'-C'}\mathbb{E}_X(t_k) - 4(K-1)\frac{\sigma^2}{\sqrt{2\pi}}\frac{e^{-\frac{1}{2}}}{1+e^{-\frac{1}{2}}}P$$

$$\geq E(t_k^2) - 4(K-1)\frac{\sigma}{\sqrt{2\pi}}e^{-\frac{1}{2}}\frac{C'+2}{D'-C'}\sqrt{E(t_k^2)} - 4(K-1)\frac{\sigma^2}{\sqrt{2\pi}}\frac{e^{-\frac{1}{2}}}{1+e^{-\frac{1}{2}}}P$$

On the other hand, we have proof $\mathbb{P}(\|Y_k - Y_k^*\|_2^2 \leq \lambda) \leq \sqrt{\frac{e\lambda}{\|\Sigma\|_{min}\|B_k^* - B_{k,0}\|_2^2}}$ in the proof of Lemma 3.1, and we know for at least probably $1 - (K-1)P$, the condition $t_k = \|Y_k - Y_k^*\|_2$ and $a > \sigma(1+2\log(\frac{D'-C'}{C'+2}))^{\frac{1}{2}}$ holds. So, under this condition, we denote $F_{k,k} = \|B_k^* - B_{k,0}\|_2$, it is easy to see if $(K-1)P < 1$ we have:

$$E(t_k^2) \geq \frac{1}{1-P(K-1)}\int_{\lambda=0}^{\frac{(1-P(K-1))^2\|\Sigma\|_{min}F_{k,k}^2}{e}}(1-P(K-1)-\sqrt{\frac{e\lambda}{\|\Sigma\|_{min}F_{k,k}^2}})d\lambda$$

$$= \frac{(1 - P(K-1))^2 \|\Sigma\|_{min} F_{k,k}^2}{3e}$$

Thus for $k$-th sub distribution, we have for probably at most $P(K-1)$, $t_k^2 - 4(K-1)\frac{\sigma}{\sqrt{2\pi}}(t_k + a)e^{-\frac{a^2}{2\sigma^2}} \geq -\frac{2\sigma^2}{\pi}(K-1)^2 - \sigma^2\sqrt{\frac{2}{\pi}}(K-1)$ and for at least probably $1 - P(K-1)$, we denote $T = (1 - P(K-1))F_{k,k}\sqrt{\frac{\|\Sigma\|_{min}}{3e}}$, the $t_k^2 - 4(K-1)\frac{\sigma}{\sqrt{2\pi}}(t_k + a)e^{-\frac{a^2}{2\sigma^2}} \geq T^2 - 4(K-1)\frac{\sigma}{\sqrt{2\pi}}e^{-\frac{1}{2}}\frac{C'+2}{D'-C'}T - 4(K-1)\frac{\sigma^2}{\sqrt{2\pi}}\frac{e^{-\frac{1}{2}}}{1+e^{-\frac{1}{2}}}P$ if $T \geq 2(K-1)\frac{\sigma}{\sqrt{2\pi}}e^{-\frac{1}{2}}\frac{C'+2}{D'-C'}$. Then we get the lower bound of $\mathbb{E}_{Y \sim N(Y_k, \sigma^2 I_q)} \min_{1 \leq k' \leq K} \|Y - Y_{k'}^*\|_2^2$:

$$\mathbb{E}_{Y \sim N(Y_k, \sigma^2 I_q)} \min_{1 \leq k' \leq K} \|Y - Y_{k'}^*\|_2^2$$

$$\geq q\sigma^2 + t_k^2 - 4(K-1)\frac{\sigma}{\sqrt{2\pi}}(t_k + a)e^{-\frac{a^2}{2\sigma^2}}$$

$$\geq q\sigma^2 - P(K-1)\left(\frac{2\sigma^2}{\pi}(K-1)^2 + 2\sigma^2\sqrt{\frac{2}{\pi}}(K-1)\right)$$

$$+(1 - P(K-1))\left(T^2 - 4(K-1)\frac{\sigma}{\sqrt{2\pi}}e^{-\frac{1}{2}}\frac{C'+2}{D'-C'}T - 4(K-1)\frac{\sigma^2}{\sqrt{2\pi}}\frac{e^{-\frac{1}{2}}}{1+e^{-\frac{1}{2}}}P\right)$$

Notice that this bound holds for any $1 \leq k \leq K$, So if for any $k$, $T > \sigma\{(K-1)\sqrt{\frac{2}{\pi}}e^{-\frac{1}{2}}\frac{C'+2}{D'-C'} + \sqrt{\frac{1}{c}\frac{2}{e\pi}(K-1)^2(\frac{C'+2}{D'-C'})^2 + \frac{1}{c}2(K-1)\sqrt{\frac{2}{\pi}}\frac{e^{-\frac{1}{2}}}{1+e^{-\frac{1}{2}}}P + \frac{1}{c}\frac{P(K-1)}{1-P(K-1)}(\frac{2}{\pi}(K-1)^2 + 2\sqrt{\frac{2}{\pi}}(K-1))}\}$, similar to the proof of theorem 3.1, we have $\mathbb{E}_{X,Y,\theta^*} m(X,Y,\theta^*) = \sum_{k=1}^K p_k \mathbb{E}_{Y \sim N(Y_k, \sigma^2 I_q)} \min_{1 \leq k' \leq K} \|Y - Y_{k'}^*\|_2^2 > q\sigma^2$

So, if $\theta^*$ is the global minimum of $\mathbb{E}_{X,Y,\theta} m(X,Y,\theta)$, we have for any $1 \leq k \leq K$, $T \leq \sigma\{(K-1)\sqrt{\frac{2}{\pi}}e^{-\frac{1}{2}}\frac{C'+2}{D'-C'} + \sqrt{\frac{1}{c}\frac{2}{e\pi}(K-1)^2(\frac{C'+2}{D'-C'})^2 + \frac{1}{c}2(K-1)\sqrt{\frac{2}{\pi}}\frac{e^{-\frac{1}{2}}}{1+e^{-\frac{1}{2}}}P + \frac{1}{c}\frac{P(K-1)}{1-P(K-1)}(\frac{2}{\pi}(K-1)^2 + 2\sqrt{\frac{2}{\pi}}(K-1))}\}$, which means $F_{k,k} \leq \frac{\sigma}{1-P(K-1)}\sqrt{\frac{3e}{\|\Sigma\|_{min}}}\{(K-1)\sqrt{\frac{2}{\pi}}e^{-\frac{1}{2}}\frac{C'+2}{D'-C'}$

$+\sqrt{\frac{1}{c}\frac{2}{e\pi}(K-1)^2(\frac{C'+2}{D'-C'})^2 + \frac{1}{c}2(K-1)\sqrt{\frac{2}{\pi}}\frac{e^{-\frac{1}{2}}}{1+e^{-\frac{1}{2}}}P + \frac{1}{c}\frac{P(K-1)}{1-P(K-1)}(\frac{2}{\pi}(K-1)^2 + 2\sqrt{\frac{2}{\pi}}(K-1))}\}$

$\square$

### A.4 PROOF OF THEOREM 3.4

*Proof.* Any one-step Gibbs sampling included in our algorithm 1 contains a sample step of $\hat{\theta}$ and a sample step of $\hat{\mathcal{U}}$. In iteration $t$, we have: $vec(\hat{B}_k^{(t)}) \sim N(vec((X_k^{\mathcal{U}\top} X_k^{\mathcal{U}} + \frac{T_t}{2\kappa}I_p)^{-1} X_k^{\mathcal{U}\top} Y_k^{\mathcal{U}}), \frac{T_t}{2}(I_q \otimes X_k^{\mathcal{U}\top} X_k^{\mathcal{U}} + \frac{T_t}{2\kappa}I_{p \times q})^{-1})$ where $X_k^{\mathcal{U}}$ is the matrix whose rows are all $x_i | \hat{u}_i^{(t)} = k$, $Y_k^{\mathcal{U}}$ is the matrix whose rows are all $y_i | \hat{u}_i^{(t)} = k$ and $\hat{\mathcal{U}}^{(t)} = (\hat{u}_1^{(t)}, \cdots, \hat{u}_n^{(t)})$ is the result of the clustering of iterations $t$. Then the result of the clustering of iterations $t$ is generated by: $p(\hat{u}_j^{(t+1)} = k) \propto \exp(-\frac{\|y_j - x_j \hat{B}_k^{(t)}\|_2^2}{T_t})$ for any $1 \leq j \leq n$

Notice that for any $1 \leq j \leq n$ and $t \geq 1$, $\hat{\mathcal{U}}_j^{(t)}$ have only $K$ different values to take. Therefore, the number of states that $\hat{\mathcal{U}}^{(t)}$ can take is finite. Based on the properties of Gibbs sampling, it is easy to see that $\hat{\mathcal{U}}^{(t)}$ itself can be regarded as a discrete-time Markov chain in finite state space $S$, and the transition probability can be written as:

$$P(\hat{\mathcal{U}}^{(t+1)} = \mathcal{U}_2 | \hat{\mathcal{U}}^{(t)} = \mathcal{U}_1)$$

$$= \int_\theta P(\hat{\mathcal{U}}^{(t+1)} = \mathcal{U}_2 | \hat{\theta}^{(t)} = \theta) p(\hat{\theta}^{(t)} = \theta | \hat{\mathcal{U}}^{(t)} = \mathcal{U}_1) d\theta$$

$$= \int_\theta (\frac{\mathcal{E}(\theta, \mathcal{U}_2, T_t)}{\sum_{\mathcal{U} \in S} \mathcal{E}(\theta, \mathcal{U}, T_t)})(\frac{\mathcal{E}(\theta, \mathcal{U}_1, T_t)}{\int_{\tilde{\theta}} \mathcal{E}(\tilde{\theta}, \mathcal{U}_1, T_t)d\tilde{\theta}})$$

$$= (\int_{\tilde{\theta}} \mathcal{E}(\tilde{\theta}, \mathcal{U}_1, T_t)d\tilde{\theta})^{-1}(\int_\theta \frac{\mathcal{E}(\theta, \mathcal{U}_2, T_t)\mathcal{E}(\theta, \mathcal{U}_1, T_t)}{\sum_{\mathcal{U} \in S} \mathcal{E}(\theta, \mathcal{U}, T_t)} d\theta)$$

The equivalent above tells us that the transition matrix of $\hat{\mathcal{U}}^{(t)}$ at each step is the transition matrix of an invertible Markov chain. Furthermore, the transition probability between any two states is nonzero for any $t$. So, the distribution:

$$P(\hat{\mathcal{U}}^{(t)} = \mathcal{U}) \propto \int_{\tilde{\theta}} \mathcal{E}(\tilde{\theta}, \mathcal{U}, T_t)d\tilde{\theta}$$

$$= \prod_{k=1}^K (\pi T_t)^{\frac{pq}{2}} \det|X_k^{\mathcal{U}\top} X_k^{\mathcal{U}} + \frac{T_t}{2\kappa} I_p|^{-\frac{q}{2}} \exp\left(-\frac{tr(Y_k^{\mathcal{U}\top} Y_k^{\mathcal{U}} - Y_k^{\mathcal{U}\top} X_k^{\mathcal{U}}(X_k^{\mathcal{U}\top} X_k^{\mathcal{U}} + \frac{T_t}{2\kappa} I_p)^{-1})X_k^{\mathcal{U}\top} Y_k^{\mathcal{U}})}{T_t}\right)$$

is the stationary distribution of the transition probability matrix of iteration $t$.$\Pi^{(t)}$

To further advance the proof, we use $\Psi^{(t)}$ denote the distribution of $\hat{\mathcal{U}}^{(t)}$, $\Pi^{(t)}$ to denote the stationary distribution corresponding to the transition probability matrix from $\hat{\mathcal{U}}^{(t)}$ to $\hat{\mathcal{U}}^{(t+1)}$. $\gamma^{(t)}$ $\gamma^{(t)}$ is the spectral gap of the transition probability matrix. Thus, we have $\|\Psi^{(t+1)} - \Pi^{(t)}\|_{TV} \le (1 - \gamma^{(t)})\|\Psi^{(t)} - \Pi^{(t)}\|_{TV}$. According to Aldous' inequality (Aldous, 2006; Levin & Peres, 2017) we have:

$$\gamma^{(t)} \ge \frac{1}{2 \max_{\mathcal{U}_1, \mathcal{U}_2} \mathbb{E}\tau_{\mathcal{U}_1, \mathcal{U}_2}} \ge \frac{\min_{\mathcal{U}_1, \mathcal{U}_2} P(\hat{\mathcal{U}}_1^{(t+1)} = \hat{\mathcal{U}}_2^{(t+1)} | \hat{\mathcal{U}}_1^{(t)} = \mathcal{U}_1, \hat{\mathcal{U}}_2^{(t)} = \mathcal{U}_2)}{2}$$

$$\ge \frac{\min_{\mathcal{U}_1, \mathcal{U}_2} \sum_{\mathcal{U} \in S} P(\mathcal{U}^{(t+1)} = \mathcal{U} | \mathcal{U}^{(t)} = \mathcal{U}_2) P(\mathcal{U}^{(t+1)} = \mathcal{U} | \mathcal{U}^{(t)} = \mathcal{U}_1)}{2}$$

$$\ge \frac{\min_{\mathcal{U}_1, \mathcal{U}_2} P(\mathcal{U}^{(t+1)} = \mathcal{U}_2 | \mathcal{U}^{(t)} = \mathcal{U}_1)}{2}$$

For any $\mathcal{U}_1, \mathcal{U}_2 \in S$, $P(\mathcal{U}^{(t+1)} = \mathcal{U}_2 | \mathcal{U}^{(t)} = \mathcal{U}_1)$ can be written as $(\int_{\tilde{\theta}} \mathcal{E}(\tilde{\theta}, \mathcal{U}_1, T_t)d\tilde{\theta})^{-1}(\int_\theta \frac{\mathcal{E}(\theta, \mathcal{U}_2, T_t)\mathcal{E}(\theta, \mathcal{U}_1, T_t)}{\sum_{\mathcal{U} \in S} \mathcal{E}(\theta, \mathcal{U}, T_t)} d\theta)$ it is obvious that $\mathcal{E}(\theta, \mathcal{U}, T_t) \le \prod_{k=1}^K \exp(-\frac{\|\hat{B}_k\|_2^2}{2\kappa})$ so we can prove that there is constant $E^*$ and $E^{**}$ which do not relate to $t$ and $T$, equivalently.

$$(\int_{\tilde{\theta}} \mathcal{E}(\tilde{\theta}, \mathcal{U}_1, T_t)d\tilde{\theta})^{-1}(\int_\theta \frac{\mathcal{E}(\theta, \mathcal{U}_2, T_t)\mathcal{E}(\theta, \mathcal{U}_1, T_t)}{\sum_{\mathcal{U} \in S} \mathcal{E}(\theta, \mathcal{U}, T_t)} d\theta)$$

$$\ge E^* \int_\theta \mathcal{E}(\theta, \mathcal{U}_2, T_t)\mathcal{E}(\theta, \mathcal{U}_1, T_t)d\theta \ge E^* \exp(-\frac{E^{**}}{T_t})$$

holds for any $\mathcal{U}_1, \mathcal{U}_2$.

So we have proved the spectral gap $\gamma^{(t)} \ge \frac{E^*}{2} \exp(-\frac{E^{**}}{T_t}) \ge \frac{E^*}{2} \exp(-\frac{E^{**}}{T_{(1)}}(\log t)^\alpha)$, then we have $\|\Psi^{(t+1)} - \Pi^{(t+1)}\|_{TV} \le \|\Psi^{(t+1)} - \Pi^{(t)}\|_{TV} + \|\Pi^{(t)} - \Pi^{(t+1)}\|_{TV} \le (1 - \gamma^{(t)})\|\Psi^{(t)} - \Pi^{(t)}\|_{TV} + \|\Pi^{(t)} - \Pi^{(t+1)}\|_{TV}$

Because in distribution $\Pi^{(t)}$, we have:

$$P(\hat{\mathcal{U}}^{(t)} = \mathcal{U}) \propto \prod_{k=1}^K (\pi T_t)^{\frac{pq}{2}} \det|X_k^{\mathcal{U}\top} X_k^{\mathcal{U}} + \frac{T_t}{2\kappa} I_p|^{-\frac{q}{2}}$$

$$\exp\left(-\frac{tr(Y_k^{\mathcal{U}\top} Y_k^{\mathcal{U}} - Y_k^{\mathcal{U}\top} X_k^{\mathcal{U}}(X_k^{\mathcal{U}\top} X_k^{\mathcal{U}} + \frac{T_t}{2\kappa} I_p)^{-1})X_k^{\mathcal{U}\top} Y_k^{\mathcal{U}})}{T_t}\right)$$

So there is constant $E^{***} > 0$, letting $|\log(\frac{P(\hat{\mathcal{U}}^{(t)} = \mathcal{U})}{P(\hat{\mathcal{U}}^{(t+1)} = \mathcal{U})})| \le E^{***}|\frac{1}{T_t} - \frac{1}{T_{t+1}}| \le \frac{E^{***}}{T_1} \frac{\alpha(\log(t))^{\alpha-1}}{t}$

So,

$$\|\Pi^{(t)} - \Pi^{(t+1)}\|_{TV} \leq \sum_{\mathcal{U} \in S, P(\hat{\mathcal{U}}^{(t)} = \mathcal{U}) > P(\hat{\mathcal{U}}^{(t+1)} = \mathcal{U})} (P(\hat{\mathcal{U}}^{(t)} = \mathcal{U}) - P(\hat{\mathcal{U}}^{(t+1)} = \mathcal{U}))$$

$$\leq (1 - \exp(-\frac{E^{***}}{T_{(1)}} \frac{\alpha(\log(t))^{\alpha-1}}{t})) \sum_{\mathcal{U} \in S, P(\hat{\mathcal{U}}^{(t)} = \mathcal{U}) > P(\hat{\mathcal{U}}^{(t+1)} = \mathcal{U})} P(\hat{\mathcal{U}}^{(t)} = \mathcal{U})$$

$$\leq (1 - \exp(-\frac{E^{***}}{T_{(1)}} \frac{\alpha(\log(t))^{\alpha-1}}{t})) \leq \frac{E^{***}}{T_{(1)}} \frac{\alpha(\log(t))^{\alpha-1}}{t}$$

So we have

$$\|\Psi^{(t+1)} - \Pi^{(t+1)}\|_{TV} \leq (1 - \gamma^{(t)})\|\Psi^{(t)} - \Pi^{(t)}\|_{TV} + \|\Pi^{(t)} - \Pi^{(t+1)}\|_{TV}$$

$$\leq (1 - \frac{E^*}{2} \exp(-\frac{E^{**}}{T_{(1)}}(\log t)^\alpha))\|\Psi^{(t)} - \Pi^{(t)}\|_{TV} + \frac{E^{***}}{T_{(1)}} \frac{\alpha(\log(t))^{\alpha-1}}{t}$$

If there is a constant $E$ let $\|\Psi^{(t)} - \Pi^{(t)}\|_{TV} \leq E \exp(-\frac{E^{**}}{T_{(1)}}(\log t)^\alpha)$, then

$$\|\Psi^{(t+1)} - \Pi^{(t+1)}\|_{TV} \leq (1 - \frac{E^*}{2} \exp(-\frac{E^{**}}{T_{(1)}}(\log t)^\alpha))E \exp(-\frac{E^{**}}{T_{(1)}}(\log t)^\alpha) + \frac{\alpha(\log(t))^{\alpha-1}}{t}$$

$$\leq E \exp(-\frac{E^{**}}{T_{(1)}}\log(t+1)^\alpha)$$

$$-\left(\frac{EE^{**}}{2}\exp(-\frac{2E^{**}}{T_{(1)}}(\log t)^\alpha) - E\exp(-\frac{E^{**}}{T_{(1)}}\log(t+1)^\alpha)\frac{\alpha(\log(t))^{\alpha-1}}{t} - \frac{E^{***}}{T_{(1)}}\frac{\alpha(\log(t))^{\alpha-1}}{t}\right)$$

When $E$ and $t$ are sufficiently large, $\frac{EE^{**}}{2}\exp(-\frac{2E^{**}}{T_{(1)}}(\log t)^\alpha) - E\exp(-\frac{E^{**}}{T_{(1)}}\log(t+1)^\alpha)\frac{\alpha(\log(t))^{\alpha-1}}{t} - \frac{E^{***}}{T_{(1)}}\frac{\alpha(\log(t))^{\alpha-1}}{t} > 0$, so we have $\|\Psi^{(t+1)} - \Pi^{(t+1)}\|_{TV} \leq E\exp(-\frac{E^{**}}{T_{(1)}}(\log t)^\alpha)$, so according to the principle of induction, we can prove that there exists $E$:

$$\|\Psi^{(t)} - \Pi^{(t)}\|_{TV} \leq E\exp(-\frac{E^{**}}{T_{(1)}}(\log t)^\alpha)$$

holds for any $t$.

Now we can analysis the $\Pi^{(t)}$, According to Assumption 1 and the properties of the distribution of $\Pi^{(t)}$. If we denote $\mathcal{E}_{\mathcal{U},k}^{(t)} = tr(Y_k^{\mathcal{U}\top}Y_k^{\mathcal{U}} - Y_k^{\mathcal{U}\top}X_k^{\mathcal{U}}(X_k^{\mathcal{U}\top}X_k^{\mathcal{U}} + \frac{T_t}{2\kappa}I_p)^{-1})X_k^{\mathcal{U}\top}Y_k^{\mathcal{U}})$, then in distribution $\Pi^{(t)}$, there is $P(\hat{\mathcal{U}}^{(t)} = \mathcal{U}) \propto \prod_{k=1}^K (\pi T_t)^{\frac{pq}{2}} \det|X_k^{\mathcal{U}\top}X_k^{\mathcal{U}} + \frac{T_t}{2\kappa}I_p|^{-\frac{q}{2}} \exp(-\frac{\mathcal{E}_{\mathcal{U},k}^{(t)}}{T_t}) = \exp(-\frac{\sum_{k=1}^K \mathcal{E}_{\mathcal{U},k}^{(t)}}{T_t}) \prod_{k=1}^K (\pi T_t)^{\frac{pq}{2}} \det|X_k^{\mathcal{U}\top}X_k^{\mathcal{U}} + \frac{T_t}{2\kappa}I_p|^{-\frac{q}{2}}$.

By Assumption 1, the sum $\sum_{k=1}^K \mathcal{E}_{\mathcal{U},k}$ attains its unique minimum, up to the permutation symmetry for $\{1, 2, \cdots, k\}$ at $\mathcal{U} = \mathcal{U}^*$. Furthermore, since $T_t \leq \frac{T_{(2)}}{(\log t)^\alpha}$, based on the properties of the exponential energy function during cooling, we know that there exist constants $E', E'^*$ such that: $P(\hat{\mathcal{U}}^{(t)} = \mathcal{U}) \geq 1 - E'^* \exp(-\frac{E'}{T_{(2)}}(\log t)^\alpha)$ in the sense of rearranging categories $1, 2, \cdots, K$,.

In summary, for the distribution $\Psi^{(t)}$, if in the sense of rearranging categories $1, 2, \cdots, K$ we have $\mathcal{U} = \mathcal{U}^*$, there exist constants $C^*, C^{**}$ such that: $P(\hat{\mathcal{U}}^{(t)} = \mathcal{U}) \geq 1 - C^* \exp(-C^{**}(\log t)^\alpha)$, in this time, under the condition $\hat{\mathcal{U}}^{(t)} = \mathcal{U}$ in the sense of rearranging categories $1, 2, \cdots, K$,, we have $P(\|\hat{B}_k^{(t)} - \hat{B}_{\pi(k)}\| < \delta) \geq 1 - C_\delta^* \exp(-C_\delta^{**}(\log t)^\alpha)$ where permutation $\pi$ of categories transfers the cluster result $\mathcal{U}^*$ to $\mathcal{U}$. That is the proof. $\qquad\square$

# B  THEOREMS OF FINITE-SAMPLE GUARANTEES FOR ESTIMATE PARAMETER $\hat{\theta}$ AND THEIR PROOF

This appendix provides a detailed finite-sample analysis of the proposed estimator. We present a series of theoretical results, including an upper bound on the estimation error between the estimator $\hat{\theta}$ and the true parameter $\theta_0$, along with the corresponding guarantees on the classification accuracy and misclassification rate. The subsequent sections present the formal statements of these theorems and their proofs.

Our theoretical approach departs from the conventional finite-sample analysis of K-means clustering, which typically requires a boundedness assumption on the observed samples (Kim & Lim, 2025). Instead, we avoid any sample-level constraints by restricting our analysis to a compact parameter space, $\Theta_M$. This constraint is not merely a theoretical convenience, but is naturally enforced by the regularization mechanism within Algorithm 1. This addresses potential optimization instabilities in the finite-sample regime, such as near-degenerate gradients. Within this well-defined framework, Lemma B.1 and Theorems B.2 and B.3 establish a non-asymptotic theory for the properties of the global minimum. The definition of the parameter space:

$$\Theta_M = \left\{ \hat{\theta} \mid \forall 1 \leq k' \leq K', \|\hat{B}_{k'}\|_F \leq M \right\} \tag{5}$$

It is important to contextualize the conditions under which these theorems hold. For the separation condition in Theorems B.2 and B.3 to be non-vacuous, the sample size $n$ must be sufficiently large (e.g., $n100K^2M^2/D^2$). The key insight from this result is that, after leaving out the model's inherent systematic bias (that is, the difference between $\theta^*$ and $\theta_0$), the intrinsic statistical uncertainty of the estimator $\hat{\theta}$ still decays at the standard parametric rate of $\mathcal{O}_p(n^{-1/2})$.

## B.1  LEMMA B.1 AND IT'S PROOF

**Theorem B.1.** *Under Assumptions 1, 2 and 3, if we have the inequality condition $M > N = \max\{\|B_{1,0}\|_F, \|B_{2,0}\|_F \cdots, \|B_{K,0}\|_F\}$ where $C = K\sqrt{3e}\{(K-1)\sqrt{\frac{2}{\pi}} + \sqrt{\frac{1}{c}((K-1)^2\frac{2}{\pi}) + 2(K-1)\sqrt{\frac{2}{\pi}}}\}$, we denote the estimator $\hat{\theta} = (\hat{B}_1, \hat{B}_2, \cdots, \hat{B}_K)$ minimize the*

$$\frac{1}{n}\sum_{i=1}^{n} m(x_i, y_i, \theta),$$

*then under condition $\hat{\theta} \in \Theta_M$ for any $1 \leq k' \leq K$ and $1 \leq k \leq K$ and for at least probability $1-t$, there exist a $1 \leq \pi(k) \leq K$ satisfy:*

$$\|B_{k,0} - \hat{B}_{\pi(k)}\|_F \leq C_{n,t} \frac{\sigma}{\sqrt{\|\Sigma\|_{min}}}$$

*where*

$$C_{n,t} = K\sqrt{3e}\{(K-1)\sqrt{\frac{2}{\pi}} + \sqrt{\frac{1}{c}\frac{2}{\pi}(K-1)^2 + \frac{1}{c}2(K-1)\sqrt{\frac{2}{\pi}} + \frac{1}{\sigma^2}(C'_n + C''_{n,t})}\}$$

*, $C'_n = \sqrt{\frac{32}{n}}K(\sqrt{q(q+2)}\sigma^2 + 2(M+N)\sigma\sqrt{\|\Sigma\|_2} + \sqrt{3}(M+N)^2\|\Sigma\|_2)$ and $C''_{n,t} = \sqrt{\frac{32}{n}}(q\sigma^2 + p(M+N)^2\|\Sigma\|_2)\log(\frac{n(p+q)+1}{t})^{\frac{3}{2}}$*

*Proof.* Let $\hat{\theta} = \{\hat{B}_1, \hat{B}_2, \cdots, \hat{B}_K\}$ be the parameter, $\Theta = \{\hat{\theta}\|\hat{B}_k\|_F < M\}$ be the parameter space. We define $R(\theta) = \mathbb{E}_{(X,Y)}[m(X,Y,\theta)]$, $R_n(\theta) = \frac{1}{n}\sum_{i=1}^{n} m(x_i, y_i, \theta)$. Then $R(\theta^*) = \min_{\theta\in\Theta} R(\theta)$ and $R_n(\hat{\theta}) = \min_{\theta\in\Theta} R_n(\theta)$, then we have $R(\hat{\theta}) - R(\theta_0) \leq R(\hat{\theta}) - R(\theta^*) \leq \sup_{\theta\in\Theta}(R_n(\theta) - R(\theta)) + \sup_{\theta\in\Theta}(R(\theta) - R_n(\theta))$.

For both $\sup_{\theta\in\Theta}(R_n(\theta) - R(\theta))$ and $\sup_{\theta\in\Theta}(R(\theta) - R_n(\theta))$, we can bound them by Rademacher complexity:

$$RC = \mathbb{E}_{x_i, y_i, \delta_i}[\sup_{\theta} |\frac{1}{n}\sum_{i=1}^{n} \delta_i m(x_i, y_i, \theta)|]$$

where $\{\delta_i\}$ is an i.i.d sequence of two-point distribution random variable satisfies $P(\delta_i = 1) = P(\delta_i = -1) = \frac{1}{2}$

According to Symmetrization Lemma, We have $E \sup_{\theta \in \Theta}(R(\theta) - R_n(\theta)) \leq 2RC$ and $E \sup_{\theta \in \Theta}(R_n(\theta) - R(\theta)) \leq 2RC$.

To give an upper bound of $RC$, we use the theorm proved by the work Maurer (2016) notice that

$$RC = \mathbb{E}_{x_i,y_i,\delta_i}[\sup_{B_1,\dots,B_K} |\frac{1}{n}\sum_{i=1}^{n} \delta_i \min_{1 \leq k \leq K} \|y_i - x_i B_k\|_2^2|]$$

$$\leq \sqrt{2}\mathbb{E}_{x_i,y_i,\delta_{ik}}[\sup_{\|B\|_F \leq M} \frac{1}{n}|\sum_{i=1}^{n}\sum_{k=1}^{K} \delta_{ik}\|y_i - x_i B\|_2^2|]$$

$$\leq \sqrt{2}K\mathbb{E}_{x_i,y_i,\delta_i}[\sup_{\|B\|_F \leq M} \frac{1}{n}|\sum_{i=1}^{n} \delta_i\|y_i - x_i B\|_2^2|]$$

$$= \sqrt{2}K\mathbb{E}_{x_i,\epsilon_i,\delta_i}[\sup_{\|B\|_F \leq M} \frac{1}{n}|\sum_{i=1}^{n} \delta_i\|\epsilon_i + x_i(B_{u_{i,0},0} - B)\|_2^2|]$$

$$\leq \frac{1}{n}\sqrt{2}K(\mathbb{E}_{x_i,\epsilon_i,\delta_i}[\sup_{\|B\|_F \leq M} |\sum_{i=1}^{n} \delta_i\|\epsilon_i\|_2^2|] + 2\mathbb{E}_{x_i,\epsilon_i,\delta_i}[\sup_{\|B\|_F \leq M} |\sum_{i=1}^{n} \delta_i x_i(B_{u_{i,0},0} - B)\epsilon_i^T|]$$

$$+\mathbb{E}_{x_i,\epsilon_i,\delta_i}[\sup_{\|B\|_F \leq M} |\sum_{i=1}^{n} \delta_i\|x_i(B_{u_{i,0},0} - B)\|_2^2|]$$

$$\leq \frac{1}{n}\sqrt{2}K(\sqrt{nq(q+2)}\sigma^2 + 2(M+N)\sigma\sqrt{n\|\Sigma\|_2} + \sqrt{3n}(M+N)^2\|\Sigma\|_2).$$

After caculating $RC$, for any $M_n > 0$, we let $\epsilon_i = \sigma a_i$ and $x_i = b_i\Sigma^{\frac{1}{2}}$, Then $a_i \sim N(0, I_q)$ and $b_i \sim N(0, I_p)$ are both the $i$th vector of an i.i.d sequence. The probability of each component of each random vector are smaller than $M_n$ is:

$$P((\cap_{1 \leq i \leq n, 1 \leq j \leq q}|a_{ij}| < M_n) \cap (\cap_{1 \leq i \leq n, 1 \leq j \leq p}|b_{ij}| < M_n)) \geq 1 - n(p+q)e^{-\frac{M_n^2}{2}}.$$

Then, under the condition of each component of each random vector $a_i$ and $b_i$ are smaller than $M_n$(we call this the bound condition below), We have $0 \leq \|y_i - x_i B\|_2^2 = \|\sigma a_i + b_i\Sigma^{\frac{1}{2}}(B_{u_{i,0},0} - B)\|_2^2 \leq 2M_n^2(q\sigma^2 + p(M+N)^2\|\Sigma\|_2)$. So the value of $\sup_{\theta \in \Theta}(R_n(\theta) - R(\theta)) - E[\sup_{\theta \in \Theta}(R_n(\theta) - R(\theta))] = \sup_{\theta \in \Theta}(\frac{1}{n}\sum_{i=1}^{n}\min_{1 \leq k \leq K}\|y_i - x_i B_k\|_2^2 - E\min_{1 \leq k \leq K}\|y_i - x_i B_k\|_2^2)$ changes by at most $\frac{2}{n}M_n^2(q\sigma^2 + p(M+N)^2\|\Sigma\|_2)$ when one of the $(x_i, y_i)$ varies. According to the McDiarmid inequality, the tail bound of $R_n$ under the bound condition satisfies:

$$P(\sup_{\theta \in \Theta}(R_n(\theta) - R(\theta)) \geq t) \leq \exp(-\frac{t^2 n}{2M_n^4(q\sigma^2 + p(M+N)^2\|\Sigma\|_2)^2}).$$

In summary, without any condition, we have:

$$P(\sup_{\theta \in \Theta}(R_n(\theta) - R(\theta)) - E[\sup_{\theta \in \Theta}(R_n(\theta) - R(\theta))] \geq t)$$

$$\leq \exp(-\frac{t^2 n}{2M_n^4(q\sigma^2 + p(M+N)^2\|\Sigma\|_2)^2}) + n(p+q)e^{-\frac{M_n^2}{2}}.$$

We let $M_n = (\frac{t^2 n}{(q\sigma^2 + p(M+N)^2\|\Sigma\|_2)^2})^{\frac{1}{6}}$, then we have:

$$P(\sup_{\theta \in \Theta}(R_n(\theta) - R(\theta)) - E[\sup_{\theta \in \Theta}(R_n(\theta) - R(\theta))] \geq t)$$

$$\leq (n(p+q)+1)\exp(-\frac{(t^2 n)^{\frac{1}{3}}}{2(q\sigma^2 + p(M+N)^2\|\Sigma\|_2)^{\frac{2}{3}}}).$$

So for at least probability $1-t$, the $\sup_{\theta\in\Theta}(R_n(\theta)-R(\theta)) - E[\sup_{\theta\in\Theta}(R_n(\theta)-R(\theta))] \leq \sqrt{\frac{8}{n}}(q\sigma^2 + 4(M+N)^2\|\Sigma\|_2)\log(\frac{n(p+q)+1}{t})^{\frac{3}{2}}$.

Similarly, for at least probability $1-t$, the $\sup_{\theta\in\Theta}(R(\theta)-R_n(\theta)) - E[\sup_{\theta\in\Theta}(R(\theta)-R_n(\theta))] \leq \sqrt{\frac{8}{n}}(q\sigma^2 + p(M+N)^2\|\Sigma\|_2)\log(\frac{n(p+q)+1}{t})^{\frac{3}{2}}$.

In summary, the $R(\hat{\theta}_n) - R(\hat{\theta})$ satisfy:

$$R(\hat{\theta}_n) - R(\hat{\theta}) \leq \sup_{\theta\in\Theta}(R_n(\theta)-R(\theta)) + \sup_{\theta\in\Theta}(R(\theta)-R_n(\theta))$$

$$\leq 4RC + \sup_{\theta\in\Theta}(R(\theta)-R_n(\theta)) - E[\sup_{\theta\in\Theta}(R(\theta)-R_n(\theta))] + \sup_{\theta\in\Theta}(R_n(\theta)-R(\theta)) - E[\sup_{\theta\in\Theta}(R_n(\theta)-R(\theta))]$$

So, for probably at least $1-t$ we have

$$sup_{\theta\in\Theta}(R_n(\theta)-R(\theta)) + \sup_{\theta\in\Theta}(R(\theta)-R_n(\theta)) \leq 4RC + \sqrt{\frac{32}{n}}(q\sigma^2 + p(M+N)^2\|\Sigma\|_2)\log(\frac{n(p+q)+1}{t})^{\frac{3}{2}}$$

$$= \sqrt{\frac{32}{n}}K(\sqrt{q(q+2)}\sigma^2 + 2(M+N)\sigma\sqrt{\|\Sigma\|_2} + \sqrt{3}(M+N)^2\|\Sigma\|_2)$$

$$+ \sqrt{\frac{32}{n}}(q\sigma^2 + p(M+N)^2\|\Sigma\|_2)\log(\frac{n(p+q)+1}{t})^{\frac{3}{2}}$$

$$= C_n' + C_{n,t}''$$

According to the proof of the theorem 3.1, if there is a $F_{\pi(k),k} > \frac{\sigma K\sqrt{3e}}{\sqrt{\|\Sigma\|_{min}}}\{(K-1)\sqrt{\frac{2}{\pi}} + \sqrt{\frac{1}{c}\frac{2}{\pi}(K-1)^2 + \frac{1}{c}2(K-1)\sqrt{\frac{2}{\pi}} + \frac{1}{\sigma^2}(C_{n,t}'' + C_n')}\}$, we have $R(\hat{\theta}_n) > q\sigma^2 + C_{n,t}'' + C_n'$ so $R(\hat{\theta}_n) - R(\hat{\theta}) > C_{n,t}'' + C_n'$. Probably for at least $1-t$ that will not happen. Thus, for probably at least $1-t$, $F_{\pi(k),k} \leq \frac{\sigma K\sqrt{3e}}{\sqrt{\|\Sigma\|_{min}}}\{(K-1)\sqrt{\frac{2}{\pi}} + \sqrt{\frac{1}{c}\frac{2}{\pi}(K-1)^2 + \frac{1}{c}2(K-1)\sqrt{\frac{2}{\pi}} + \frac{1}{\sigma^2}(C_{n,t}'' + C_n')}\}$. That ends the proof. $\qquad\square$

### B.2 THEOREM B.2 AND IT'S PROOF

**Theorem B.2.** *We let $i = \arg\min_{1\leq k\leq K}\|y_i - x_i\hat{B}_k\|_2^2$ and $D' = \frac{\sqrt{\|\Sigma\|_{min}}}{\sigma}D$. Under Assumptions 1, 2 and 3, if $K = K, D' > 2\sqrt{\frac{\|\Sigma\|_2}{\|\Sigma\|_{min}}}C_{n,t} + 2\sqrt{q}$ and $M > N = \max\{\|B_{1,0}\|_F, \|B_{2,0}\|_F\cdots, \|B_{K,0}\|_F\}$ where $C = K\sqrt{3e}\{(K-1)\sqrt{\frac{2}{\pi}} + \sqrt{\frac{1}{c}((K-1)^2\frac{2}{\pi}) + 2(K-1)\sqrt{\frac{2}{\pi}}}\}$, estimator $\hat{\theta} = (\hat{B}_1, \hat{B}_2, \cdots, \hat{B}_K) \in \Theta_M$ minimize the*

$$\frac{1}{n}\sum_{i=1}^{n}m(x_i, y_i, \theta).$$

*It is reasonable to assume that $k = \arg\min_{1\leq k'\leq k}\|\hat{B}_{k'} - B_{k,0}\|_F$, the estimate cluster of $(x_i, y_i)$ should be $\hat{u}_i = \min_{1\leq k\leq K}\|y_i - x_i\hat{B}_k\|_2$. If we denote $C' = C_{n,t}\sqrt{\frac{\|\Sigma\|_2}{\|\Sigma\|_{min}}}$ $\lambda = (K-1)\{\frac{C'+2\sqrt{q}}{D'-C'}(1 + 2\log(\frac{D'-C'}{C'+2\sqrt{q}}))^{\frac{1}{2}} + (\frac{C'+2\sqrt{q}}{D'-C'}(1 + 2\log(\frac{D'-C'}{C'+2\sqrt{q}}))^{\frac{1}{2}})^q + e^{\frac{1}{2}}(\frac{C'+2\sqrt{q}}{D'-C'})(1 + 2\log(\frac{D'-C'}{C'+2\sqrt{q}}))^{\frac{1}{2}}\}$ and $t_s = \frac{n^n}{s^s(n-s)^{n-s}}\lambda^s(1-\lambda)^{n-s}$ where $0 \leq s \leq n$, then for probability at least $1 - t - t_s$*

$$\sum_{i=1}^{n} I(\hat{u}_i \neq u_i) < s.$$

*Proof.* Using Lemma B.1, it is easy to find that there is at least probably $1 - t$, $\max_{1 \leq k \leq K} \|\hat{B}_k - B_{k,0}\|_F \leq C' \frac{\sigma}{\sqrt{\|\Sigma\|_2}}$.

If we have conditions $\max_{1 \leq k \leq K} \|\hat{B}_k - B_{k,0}\|_F \leq C' \frac{\sigma}{\sqrt{\|\Sigma\|_2}}$, according to lemma **??**, for any $1 \leq i \leq n$, $P(\hat{u}_i \neq u_i) \leq \lambda$. And it is easy to find after knowing the value $\hat{B}_k$, the sequence of events $\{\hat{u}_i \neq u_i\}$ is an i.i.d. sequence. So we can get the Chernoff Bound of the $P(\sum_{i=1}^{n} I(\hat{u}_i \neq u_i) \geq s)$:

$$P(\sum_{i=1}^{n} I(\hat{u}_i \neq u_i) \geq s)$$

$$\leq e^{-ts} \{\mathbb{E} e^{I(\hat{u}_i \neq u_i)}\}^n$$

$$\leq e^{-ts}(\lambda e^t + 1 - \lambda)^n = (\lambda e^{t(\frac{n-s}{n})} + (1-\lambda)e^{-t\frac{s}{n}})^n$$

We take $t = \log(\frac{s(1-\lambda)}{(n-s)\lambda})$ Then we have:

$$P(\sum_{i=1}^{n} I(\hat{u}_i \neq u_i) \geq s)$$

$$\leq (\lambda(\frac{s(1-\lambda)}{(n-s)\lambda})^{\frac{n-s}{n}} + (1-\lambda)(\frac{s(1-\lambda)}{(n-s)\lambda})^{\frac{-s}{n}})^n$$

$$= (\frac{n}{n-s}(1-\lambda)(\frac{s(1-\lambda)}{(n-s)\lambda})^{\frac{-s}{n}})^n$$

$$= \frac{n^n}{(n-s)^{n-s} s^s} \lambda^s (1-\lambda)^{n-s} = t_s$$

The above analysis is the conditional probability obtained under the condition $\max_{1 \leq k \leq K} \|\hat{B}_k - B_{k,0}\|_F \leq C'$. Since the condition $\max_{1 \leq k \leq K} |\hat{B}_k - B_{k,0}\|_F \leq C'$ is greater than $1 - t$, it follows that the probability of $P(\sum_{i=1}^{n} I(\hat{u}_i \neq u_i) < s)$ is greater than $1 - t - t_s$.

$\square$

## B.3 THEOREM B.3 AND IT'S PROOF

**Theorem B.3.** *If $D' = \frac{\sqrt{\|\Sigma\|_{min}}}{\sigma} D$. Then Under Assumptions 1, 2 and 3. If $D' > 2C_{n,t} + 2$ and estimator $\hat{\theta} = (\hat{B}_1, \hat{B}_2, \cdots, \hat{B}_K) \in \Theta_M$ minimize the*

$$\frac{1}{n} \sum_{i=1}^{n} m(x_i, y_i, \theta),$$

*Then for at least probability $1 - t$, there is a constant $C_D$, for any $1 \leq k \leq K$, there exist a $\pi(k)$ satisfy*

$$\|B_{k,0} - \hat{B}_{\pi(k)}\|_F \leq C_D \frac{\sigma}{\sqrt{\|\Sigma\|_{min}}}$$

*for any $1 \leq k \leq K$, where*

$$C_D = \frac{\sqrt{3e}}{1 - P(K-1)} \{(K-1)\sqrt{\frac{2}{\pi}} e^{-\frac{1}{2}} \frac{C'+2}{D'-C'}$$

$$+ \sqrt{\frac{1}{c} \frac{2}{e\pi}(K-1)^2(\frac{C'+2}{D'-C'})^2 + \frac{1}{c}2(K-1)\sqrt{\frac{2}{\pi}} \frac{e^{-\frac{1}{2}}}{1+e^{-\frac{1}{2}}} P + \frac{1}{c} \frac{P(K-1)}{1-P(K-1)}(\frac{2}{\pi}(K-1)^2 + 2\sqrt{\frac{2}{\pi}}(K-1)) + \frac{1}{\sigma^2}(C'_n + C''_{n,t})\}}$$

*and*

$$P = (1 + e^{\frac{1}{2}}) \frac{C'+2}{D'-C'}(1 + 2\log(\frac{D'-C'}{C'+2}))^{\frac{1}{2}}$$

*and* $C' = C_{n,t}\sqrt{\frac{\|\Sigma\|_2}{\|\Sigma\|_{min}}}$, $C'_n = \sqrt{\frac{32}{n}}K(\sqrt{q(q+2)}\sigma^2 + 2(M+N)\sigma\sqrt{\|\Sigma\|_2} + \sqrt{3}(M+N)^2\|\Sigma\|_2)$

*and* $C''_{n,t} = \sqrt{\frac{32}{n}(q\sigma^2 + p(M+N)^2\|\Sigma\|_2)\log(\frac{n(p+q)+1}{t})^{\frac{3}{2}}}$

*Proof.* This proof is similar to the proof of Theorem 3.3, for at least probability $1 - t$, we have $\min_{1 \le i < j \le k} \|B_{i,0} - B_{j,0}\|_F = D = D'\frac{\sigma}{\sqrt{\|\Sigma\|_{min}}}$, $\|B_{k,0} - \hat{B}_k\|_F = \max_{1 \le k \le K} \|B_{k,0} - \hat{B}_k\|_F \le C_{n,t}\frac{\sigma}{\sqrt{\|\Sigma\|_{min}}} = C'\frac{\sigma}{\sqrt{\|\Sigma\|_2}}$ and $D' > 2C' + 2$.

Then, we use the definition in the proof of Lemma 3.1 and B.1, denote $\hat{Y}_k = X\hat{B}_k$ and $Y_k = XB_{k,0}$ for $1 \le k \le K$, $t_k = \min_{1 \le k' \le K} \|Y_k - \hat{Y}_{k'}\|_2$ for $X$. With the condition we have, for any $k' \ne k$ we can proof:

$$\mathbb{P}\left(\|Y_{k'} - \hat{Y}_k\|_2 - \|Y_k - \hat{Y}_k\|_2 < 2\sigma(1 + 2\log(\frac{D' - C'}{C' + 2}))^{\frac{1}{2}}\right)$$

$$= \mathbb{P}\left(\|X(B_{k,0} - \hat{B}_{k'})\|_2 - \|X(B_{k,0} - \hat{B}_k)\|_2 < 2\sigma(1 + 2\log(\frac{D' - C'}{C' + 2}))^{\frac{1}{2}}\right)$$

$$\le \mathbb{P}\left(\|X(B_{k,0} - \hat{B}_k)\|_2^2 > C'^2\sigma^2(1 + 2\log(\frac{D' - C'}{C' + 2}))\right)$$

$$+ \mathbb{P}\left(\|X(B_{k,0} - \hat{B}_{k'})\|_2^2 < (C' + 2)^2\sigma^2(1 + 2\log(\frac{D' - C'}{C' + 2}))\right)$$

$$\le \frac{C' + 2}{D' - C'}(1 + 2\log(\frac{D' - C'}{C' + 2}))^{\frac{1}{2}} + e^{\frac{1}{2}}\frac{C' + 2}{D' - C'}(1 + 2\log(\frac{D' - C'}{C' + 2}))^{\frac{1}{2}}$$

$$= (1 + e^{\frac{1}{2}})\frac{C' + 2}{D' - C'}(1 + 2\log(\frac{D' - C'}{C' + 2}))^{\frac{1}{2}}$$

So, the probability of $t_k = \|Y_k - \hat{Y}_k\|_2$ could be bound, and for any $k' \ne k, t'_k = \|Y_k - \hat{Y}_{k'}\|_2, a = \frac{t'_k - t_k}{2}$ we have:

$$\mathbb{P}\left(t_k = \|Y_k - \hat{Y}_k\|_2, \text{ and } a > \sigma(1 + 2\log(\frac{D' - C'}{C' + 2}))^{\frac{1}{2}} \text{ for any } k' \ne k\right)$$

$$\ge 1 - (K - 1)(1 + e^{\frac{1}{2}})\frac{C' + 2}{D' - C'}(1 + 2\log(\frac{D' - C'}{C' + 2}))^{\frac{1}{2}}$$

We denote $P = (1 + e^{\frac{1}{2}})\frac{C'+2}{D'-C'}(1 + 2\log(\frac{D'-C'}{C'+2}))^{\frac{1}{2}}$ and reuse the conclusion obtained in the proof of Lemma 3.1. The global minimum of $\mathbb{E}_{X,Y}(X, Y, \theta)$ satisfy:

$$\mathbb{E}_{X,Y}(X, Y, \hat{\theta})$$

$$= \mathbb{E}_X \sum_{k=1}^K p_k \mathbb{E}_{Y \sim N(Y_k, \sigma^2 I_q)} \min_{1 \le k' \le K} \|Y - \hat{Y}_{k'}\|_2^2$$

$$\ge \mathbb{E}_X \sum_{k=1}^K p_k \mathbb{E}_{Y \sim N(Y_k, \sigma^2 I_q)}(\|Y - \hat{Y}_k\|_2^2 - \sum_{k' \ne k} \max\{0, \|Y - \hat{Y}_k\|_2^2 - \|Y - \hat{Y}_{k'}\|_2^2\})$$

$$\ge \mathbb{E}_X \sum_{k=1}^K p_k(q\sigma^2 + t_k^2 - 4(K - 1)\frac{\sigma}{\sqrt{2\pi}}(t_k + a)e^{-\frac{a^2}{2\sigma^2}})$$

For any $k$, for at most probability $(K - 1)P$, $t_k \ne \|Y_k - \hat{Y}_k\|_2$ or $a \le \sigma(1 + 2\log(\frac{D'-C'}{C'+2}))^{\frac{1}{2}}$, so $t_k^2 - 4(K-1)\frac{\sigma}{\sqrt{2\pi}}(t_k + a)e^{-\frac{a^2}{2\sigma^2}} \ge t_k^2 - 4\frac{\sigma}{\sqrt{2\pi}}(K-1)t_k - 4\frac{\sigma^2}{\sqrt{2\pi}}(K-1) \ge -\frac{2\sigma^2}{\pi}(K-1)^2 - \sigma^2\sqrt{\frac{2}{\pi}}(K-1)$. Otherwise, we have $t_k = \|Y_k - \hat{Y}_k\|_2$ and $a > \sigma(1 + 2\log(\frac{D'-C'}{C'+2}))^{\frac{1}{2}}$, In this case, we have:

$$t_k^2 - 4(K - 1)\frac{\sigma}{\sqrt{2\pi}}(t_k + a)e^{-\frac{a^2}{2\sigma^2}}$$

$$\geq t_k^2 - 4(K-1)\frac{\sigma}{\sqrt{2\pi}}e^{-\frac{1}{2}}\frac{C'+2}{D'-C'}t_k - 4(K-1)\frac{\sigma^2}{\sqrt{2\pi}}e^{-\frac{1}{2}}\frac{C'+2}{D'-C'}(1+2\log(\frac{D'-C'}{C'+2}))^{\frac{1}{2}}$$

$$= t_k^2 - 4(K-1)\frac{\sigma}{\sqrt{2\pi}}e^{-\frac{1}{2}}\frac{C'+2}{D'-C'}t_k - 4(K-1)\frac{\sigma^2}{\sqrt{2\pi}}\frac{e^{-\frac{1}{2}}}{1+e^{-\frac{1}{2}}}P$$

Using Jensen Inequality, under the condition of $t_k = \|Y_k - \hat{Y}_k\|_2$ and $a > \sigma(1+2\log(\frac{D'-C'}{C'+2}))^{\frac{1}{2}}$, we have

$$\mathbb{E}_X[t_k^2 - 4(K-1)\frac{\sigma}{\sqrt{2\pi}}e^{-\frac{1}{2}}\frac{C'+2}{D'-C'}t_k - 4(K-1)\frac{\sigma}{\sqrt{2\pi}}\frac{e^{-\frac{1}{2}}}{1+e^{-\frac{1}{2}}}P]$$

$$\geq E(t_k^2) - 4(K-1)\frac{\sigma}{\sqrt{2\pi}}e^{-\frac{1}{2}}\frac{C'+2}{D'-C'}\mathbb{E}_X(t_k) - 4(K-1)\frac{\sigma^2}{\sqrt{2\pi}}\frac{e^{-\frac{1}{2}}}{1+e^{-\frac{1}{2}}}P$$

$$\geq E(t_k^2) - 4(K-1)\frac{\sigma}{\sqrt{2\pi}}e^{-\frac{1}{2}}\frac{C'+2}{D'-C'}\sqrt{E(t_k^2)} - 4(K-1)\frac{\sigma^2}{\sqrt{2\pi}}\frac{e^{-\frac{1}{2}}}{1+e^{-\frac{1}{2}}}P$$

On the other hand, we have proof $\mathbb{P}(\|Y_k - \hat{Y}_k\|_2^2 \leq \lambda) \leq \sqrt{\frac{e\lambda}{\|\Sigma\|_{min}\|\hat{B}_k - B_{k,0}\|_2^2}}$ in the proof of Lemma 3.1, and we know for at least probably $1 - (K-1)P$, the condition $t_k = \|Y_k - \hat{Y}_k\|_2$ and $a > \sigma(1+2\log(\frac{D'-C'}{C'+2}))^{\frac{1}{2}}$ holds. So, under this condition, we denote $F_{k,k} = \|\hat{B}_k - B_{k,0}\|_2$, it is easy to see if $(K-1)P < 1$ we have:

$$E(t_k^2) \geq \frac{1}{1-P(K-1)}\int_{\lambda=0}^{\frac{(1-P(K-1))^2\|\Sigma\|_{min}F_{k,k}^2}{e}}(1-P(K-1)-\sqrt{\frac{e\lambda}{\|\Sigma\|_{min}F_{k,k}^2}})d\lambda$$

$$= \frac{(1-P(K-1))^2\|\Sigma\|_{min}F_{k,k}^2}{3e}$$

Thus for $k$-th sub distribution, we have for probably at most $P(K-1)$, $t_k^2 - 4(K-1)\frac{\sigma}{\sqrt{2\pi}}(t_k + a)e^{-\frac{a^2}{2\sigma^2}} \geq -\frac{2\sigma^2}{\pi}(K-1)^2 - \sigma^2\sqrt{\frac{2}{\pi}}(K-1)$ and for at least probably $1 - P(K-1)$, we denote $T = (1-P(K-1))F_{k,k}\sqrt{\frac{\|\Sigma\|_{min}}{3e}}$, the $t_k^2 - 4(K-1)\frac{\sigma}{\sqrt{2\pi}}(t_k+a)e^{-\frac{a^2}{2\sigma^2}} \geq T^2 - 4(K-1)\frac{\sigma}{\sqrt{2\pi}}e^{-\frac{1}{2}}\frac{C'+2}{D'-C'}T - 4(K-1)\frac{\sigma^2}{\sqrt{2\pi}}\frac{e^{-\frac{1}{2}}}{1+e^{-\frac{1}{2}}}P$ if $T \geq 2(K-1)\frac{\sigma}{\sqrt{2\pi}}e^{-\frac{1}{2}}\frac{C'+2}{D'-C'}$. Then we get the lower bound of $\mathbb{E}_{Y\sim N(Y_k,\sigma^2 I_q)}\min_{1\leq k'\leq K}\|Y-\hat{Y}_{k'}\|_2^2$:

$$\mathbb{E}_{Y\sim N(Y_k,\sigma^2 I_q)}\min_{1\leq k'\leq K}\|Y-\hat{Y}_{k'}\|_2^2$$

$$\geq q\sigma^2 + t_k^2 - 4(K-1)\frac{\sigma}{\sqrt{2\pi}}(t_k+a)e^{-\frac{a^2}{2\sigma^2}}$$

$$\geq q\sigma^2 - P(K-1)\left(\frac{2\sigma^2}{\pi}(K-1)^2 + 2\sigma^2\sqrt{\frac{2}{\pi}}(K-1)\right)$$

$$+(1-P(K-1))\left(T^2 - 4(K-1)\frac{\sigma}{\sqrt{2\pi}}e^{-\frac{1}{2}}\frac{C'+2}{D'-C'}T - 4(K-1)\frac{\sigma^2}{\sqrt{2\pi}}\frac{e^{-\frac{1}{2}}}{1+e^{-\frac{1}{2}}}P\right)$$

Notice that this bound holds for any $1 \leq k \leq K$, So if for any $k$, $T > \sigma\{(K-1)\sqrt{\frac{2}{\pi}}e^{-\frac{1}{2}}\frac{C'+2}{D'-C'} + \sqrt{\frac{1}{c}\frac{2}{e\pi}(K-1)^2(\frac{C'+2}{D'-C'})^2 + \frac{1}{c}2(K-1)\sqrt{\frac{2}{\pi}}\frac{e^{-\frac{1}{2}}}{1+e^{-\frac{1}{2}}}P + \frac{1}{c}\frac{P(K-1)}{1-P(K-1)}(\frac{2}{\pi}(K-1)^2+2\sqrt{\frac{2}{\pi}}(K-1))}\}$, similar to the proof of theorem 3.1, we have $\mathbb{E}_{X,Y,\hat{\theta}}m(X,Y,\hat{\theta}) = \sum_{k=1}^K p_k\mathbb{E}_{Y\sim N(Y_k,\sigma^2 I_q)}\min_{1\leq k'\leq K}\|Y-\hat{Y}_{k'}\|_2^2 > q\sigma^2$

So, if $\hat{\theta}$ is the global minimum of $\mathbb{E}_{X,Y,\theta}m(X,Y,\theta)$, we have for any $1 \leq k \leq K$, $T \leq \sigma\{(K-1)\sqrt{\frac{2}{\pi}}e^{-\frac{1}{2}}\frac{C'+2}{D'-C'} + \sqrt{\frac{1}{c}\frac{2}{e\pi}(K-1)^2(\frac{C'+2}{D'-C'})^2 + \frac{1}{c}2(K-1)\sqrt{\frac{2}{\pi}}\frac{e^{-\frac{1}{2}}}{1+e^{-\frac{1}{2}}}P + \frac{1}{c}\frac{P(K-1)}{1-P(K-1)}(\frac{2}{\pi}(K-1)^2+2\sqrt{\frac{2}{\pi}}(K-1))}\}$,

which means $F_{k,k} \leq \frac{\sigma}{1-P(K-1)} \sqrt{\frac{3e}{\|\Sigma\|_{min}}} \{(K-1)\sqrt{\frac{2}{\pi}} e^{-\frac{1}{2}} \frac{C'+2}{D'-C'}$

$+ \sqrt{\frac{1}{c}\frac{2}{e\pi}(K-1)^2(\frac{C'+2}{D'-C'})^2 + \frac{1}{c}2(K-1)\sqrt{\frac{2}{\pi}}\frac{e^{-\frac{1}{2}}}{1+e^{-\frac{1}{2}}}P + \frac{1}{c}\frac{P(K-1)}{1-P(K-1)}(\frac{2}{\pi}(K-1)^2 + 2\sqrt{\frac{2}{\pi}}(K-1))}$

$\square$

## C   FIGURES OF CLASSIFICATION ACCURACY AND WCSS OF FOUR DIFFERENT ESTIMATION METHODS UNDER 16 PARAMETERS CONDITIONS IN BOTH THE TRAINING SET AND THE TESTING SET

This appendix records figures for the classification accuracy and the WCSS function in the training set and the testing set.

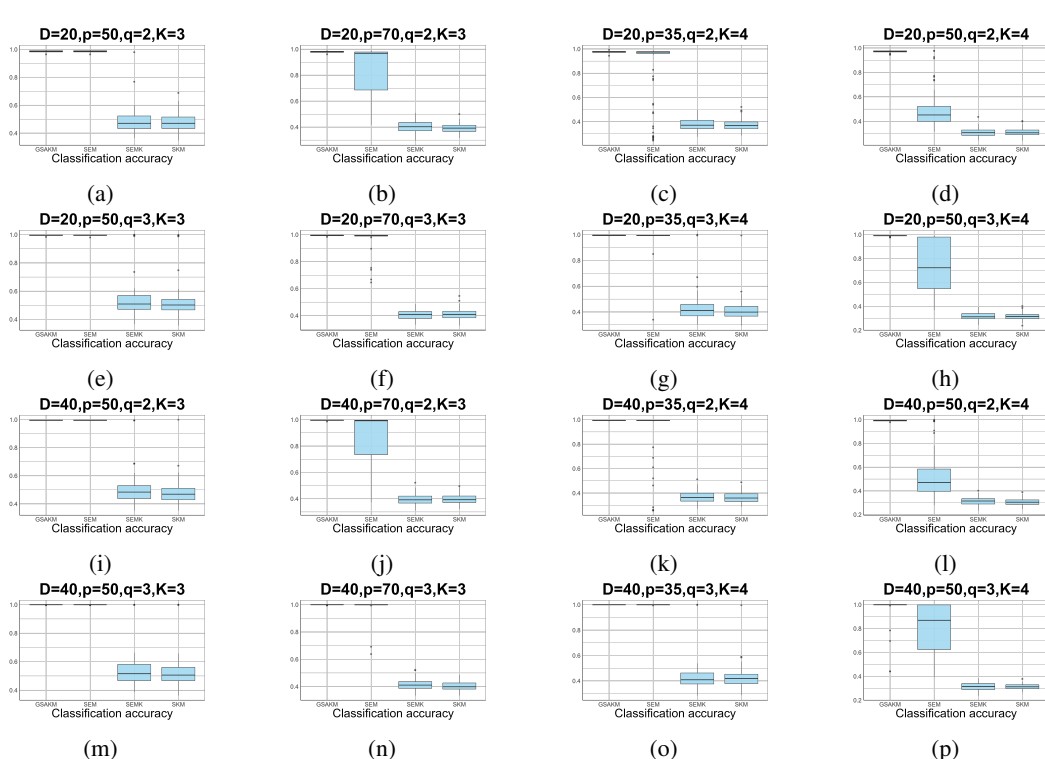

Figure 2: The box-plot of classification accuracy of four different estimation methods under 16 parameter conditions.

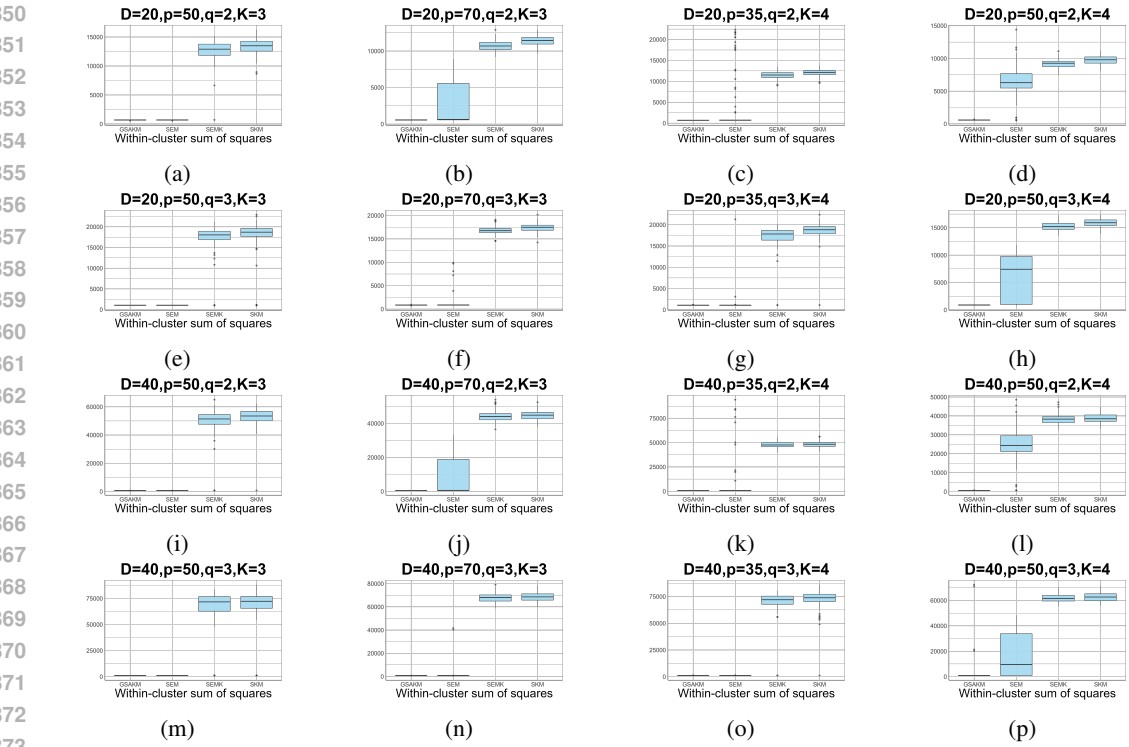

Figure 3: The box-plot of WCSS of four different estimation methods under 16 parameter conditions.

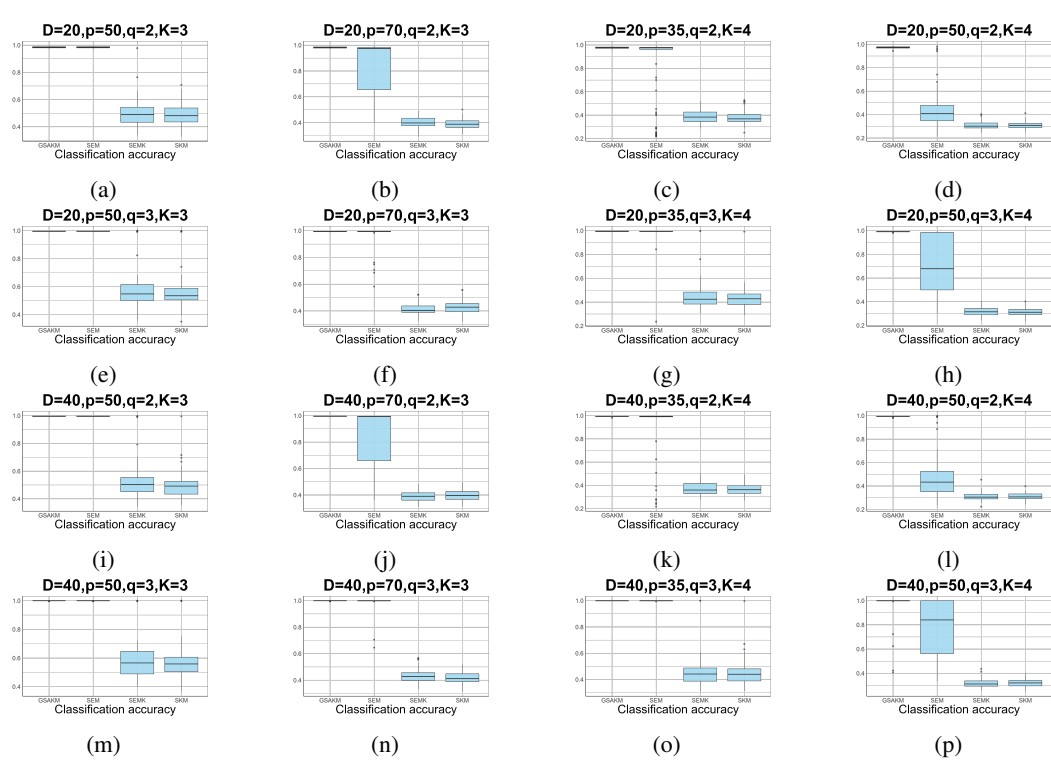

Figure 4: The box-plot of classification accuracy of four different estimation methods under 16 parameter conditions in testing set.

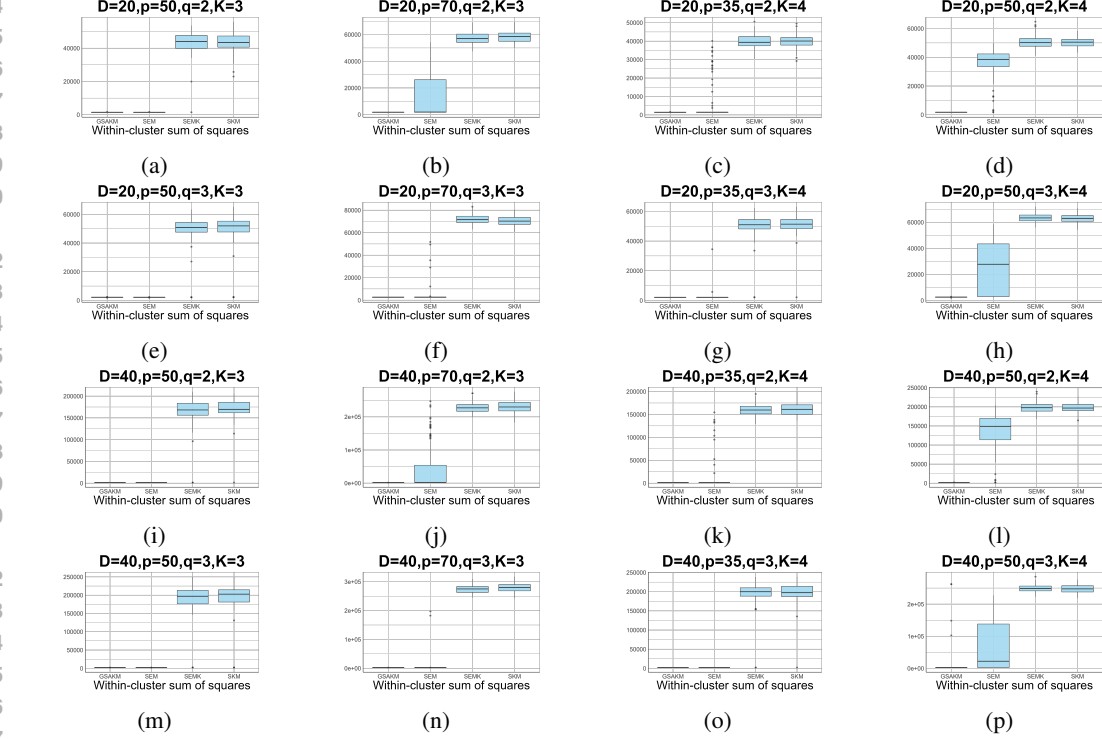

Figure 5: The box-plot of WCSS of four different estimation methods under 16 parameter conditions in testing set.

## D  PERFORMANCE COMPARISON TABLES OF OUR SIMULATION RESULTS

This appendix presents the five performance comparison tables, which compare the three different metrics used in the training set and the testing set. Results are presented as Mean (Standard Deviation) of 100 replicates.

Table 1: Performance Comparison Table of estimation errors of four different estimation methods under 16 parameter conditions.

|  |  | $P = 50, K = 3$ | $P = 70, K = 3$ | $P = 35, K = 4$ | $P = 50, K = 4$ |
|---|---|---|---|---|---|
| $D = 20$ | GSAKM | 1.15(0.095) | 1.51(0.12) | 1.16(0.11) | 1.58(0.14) |
| $, q = 2$ | SEM | 1.15(0.094) | 6.14(6.26) | 4.02(6.09) | 17.93(5.29) |
|  | SEMK | 16.63(2.29) | 19.73(1.67) | 18.98(1.86) | 21.73(2.20) |
|  | SKM | 16.42(1.86) | 20.06(1.42) | 18.62(1.86) | 21.66(2.19) |
| $D = 20$ | GSAKM | 1.36(0.087) | 1.79(0.12) | 1.34(0.10) | 1.80(0.15) |
| $, q = 3$ | SEM | 1.36(0.090) | 2.32(2.42) | 1.51(1.23) | 11.09(7.21) |
|  | SEMK | 15.09(4.40) | 19.69(1.40) | 18.31(3.10) | 21.20(2.03) |
|  | SKM | 15.42(4.24) | 19.45(1.54) | 18.15(2.51) | 21.24(1.71) |
| $D = 40$ | GSAKM | 1.12(0.082) | 1.45(0.11) | 1.11(0.089) | 1.47(0.12) |
| $, q = 2$ | SEM | 1.12(0.082) | 9.69(13.11) | 4.14(9.86) | 33.06(12.62) |
|  | SEMK | 31.94(6.73) | 39.85(3.19) | 38.05(4.27) | 43.33(4.84) |
|  | SKM | 32.71(6.95) | 39.47(3.07) | 37.63(3.30) | 42.32(4.23) |
| $D = 40$ | GSAKM | 1.34(0.084) | 1.76(0.11) | 1.33(0.10) | 4.30(12.76) |
| $, q = 3$ | SEM | 1.34(0.084) | 2.33(4.01) | 1.33(0.10) | 17.36(16.87) |
|  | SEMK | 28.28(10.37) | 38.24(2.86) | 36.18(5.33) | 42.38(3.45) |
|  | SKM | 29.22(9.34) | 38.69(2.67) | 36.60(6.45) | 42.76(3.75) |

Table 2: Performance Comparison Table of 100 times the classification accuracy of four different estimation methods under 16 parameter conditions.

|  |  | $P = 50, K = 3$ | $P = 70, K = 3$ | $P = 35, K = 4$ | $P = 50, K = 4$ |
|---|---|---|---|---|---|
| $D = 20$ | GSAKM | 98.51(0.59) | 98.04(0.67) | 97.64(0.72) | 97.11(0.78) |
| $, q = 2$ | SEM | 98.51(0.58) | 84.72(17.32) | 85.91(24.44) | 48.77(13.59) |
|  | SEMK | 48.17(8.40) | 40.40(4.40) | 37.53(5.01) | 30.89(3.19) |
|  | SKM | 47.66(6.17) | 39.15(3.65) | 37.25(4.96) | 30.92(3.29) |
| $D = 20$ | GSAKM | 99.56(0.29) | 99.49(0.33) | 99.47(0.32) | 99.13(0.42) |
| $, q = 3$ | SEM | 99.57(0.30) | 98.06(5.88) | 98.63(6.69) | 73.35(20.73) |
|  | SEMK | 55.03(14.57) | 40.81(3.66) | 42.95(10.42) | 31.43(2.87) |
|  | SKM | 53.32(13.89) | 41.03(3.92) | 41.30(8.06) | 31.40(2.93) |
| $D = 40$ | GSAKM | 99.62(0.27) | 99.50(0.30) | 99.41(0.33) | 99.20(0.46) |
| $, q = 2$ | SEM | 99.62(0.28) | 87.60(18.54) | 93.85(17.85) | 52.49(18.58) |
|  | SEMK | 50.13(10.89) | 39.50(3.86) | 36.62(4.58) | 31.34(2.98) |
|  | SKM | 47.81(8.10) | 39.67(3.63) | 36.33(4.47) | 30.79(2.82) |
| $D = 40$ | GSAKM | 99.96(0.088) | 99.94(0.12) | 99.92(0.11) | 98.26(8.61) |
| $, q = 3$ | SEM | 99.95(0.094) | 99.26(4.72) | 99.92(0.12) | 80.85(20.32) |
|  | SEMK | 56.87(17.08) | 41.34(3.89) | 42.60(10.05) | 31.36(3.24) |
|  | SKM | 54.46(15.49) | 40.39(3.55) | 42.61(8.38) | 31.33(2.34) |

Table 3: Performance Comparison Table of WCSS of four different estimation methods under 16 parameter conditions.

|  |  | $P = 50, K = 3$ | $P = 70, K = 3$ | $P = 35, K = 4$ | $P = 50, K = 4$ |
|---|---|---|---|---|---|
| $D = 20$ | GSAKM | 681.68(35.70) | 565.29(34.11) | 695.62(35.10) | 582.36(32.17) |
| $, q = 2$ | SEM | 682.84(35.77) | 2560.03(2624.32) | 3643.55(6534.88) | 6526.78(2216.05) |
|  | SEMK | 12627.42(1850.99) | 10741.43(719.03) | 11481.15(943.91) | 9201.12(641.44) |
|  | SKM | 13227.79(1414.66) | 11390.24(3.65) | 12114.49(842.06) | 9808.03(591.55) |
| $D = 20$ | GSAKM | 1046.34(44.81) | 895.76(42.03) | 1081.15(42.08) | 894.00(46.38) |
| $, q = 3$ | SEM | 1047.12(44.77) | 1214.21(1592.39) | 1304.73(2027.40) | 5935.86(3922.34) |
|  | SEMK | 16626.17(4939.88) | 16799.62(863.66) | 17195.59(2799.57) | 15231.85(799.41) |
|  | SKM | 17501.23(4822.07) | 17412.95(950.60) | 18500.90(2235.11) | 15920.06(715.56) |
| $D = 40$ | GSAKM | 692.32(36.28) | 578.18(34.86) | 711.20(35.88) | 599.59(33.91) |
| $, q = 2$ | SEM | 692.60(36.26) | 7628.15(11016.12) | 6241.16(19028.99) | 23775.27(9879.18) |
|  | SEMK | 49458.63(10240.71) | 44304.19(3281.83) | 47640.71(3198.72) | 38230.18(2775.92) |
|  | SKM | 52573.70(6958.00) | 44796.48(2906.76) | 48313.04(3247.59) | 38806.31(2460.36) |
| $D = 40$ | GSAKM | 1050.82(44.87) | 864.53(42.16) | 1086.49(42.64) | 2713.22(10293.99) |
| $, q = 3$ | SEM | 1050.90(44.91) | 1666.65(5641.26) | 1086.63(42.71) | 17051.68(17027.03) |
|  | SEMK | 63092.95(24163.18) | 67777.92(3650.69) | 70209.06(11271.25) | 61858.14(2939.07) |
|  | SKM | 66013.13(21619.79) | 68708.04(3899.46) | 72019.29(9680.24) | 62615.94(3398.19) |

Table 4: Performance Comparison Table of 100 times the classification accuracy of four different estimation methods under 16 parameter conditions in testing set.

| | | $P = 50, K = 3$ | $P = 70, K = 3$ | $P = 35, K = 4$ | $P = 50, K = 4$ |
|---|---|---|---|---|---|
| $D = 20$ | GSAKM | 98.35(0.59) | 98.24(0.56) | 97.67(0.67) | 97.28(0.73) |
| , $q = 2$ | SEM | 98.36(0.60) | 83.23(19.38) | 84.89(26.21) | 43.99(14.52) |
| | SEMK | 49.58(8.77) | 40.33(4.18) | 37.74(5.23) | 30.78(3.26) |
| | SKM | 49.44(7.68) | 38.91(3.84) | 38.65(5.15) | 31.10(2.99) |
| $D = 20$ | GSAKM | 99.59(0.28) | 99.42(0.33) | 99.45(0.36) | 99.19(0.39) |
| , $q = 3$ | SEM | 99.57(0.28) | 97.91(6.67) | 98.50(7.72) | 70.66(22.57) |
| | SEMK | 58.34(14.23) | 41.31(4.30) | 44.35(11.09) | 31.74(3.31) |
| | SKM | 56.67(13.42) | 42.64(4.83) | 43.40(8.46) | 31.48(3.33) |
| $D = 40$ | GSAKM | 99.56(0.31) | 99.54(0.27) | 99.35(0.33) | 99.35(0.40) |
| , $q = 2$ | SEM | 99.55(0.31) | 85.96(20.76) | 93.34(19.03) | 48.42(19.88) |
| | SEMK | 51.66(11.39) | 39.11(3.85) | 37.07(5.49) | 30.98(3.28) |
| | SKM | 49.52(8.79) | 40.08(3.78) | 36.51(4.58) | 30.94(2.86) |
| $D = 40$ | GSAKM | 99.95(0.10) | 99.94(0.11) | 99.93(0.12) | 98.07(9.36) |
| , $q = 3$ | SEM | 99.95(0.11) | 99.29(4.57) | 99.93(0.13) | 77.51(23.19) |
| | SEMK | 60.56(16.40) | 42.90(4.74) | 45.10(10.45) | 31.49(3.34) |
| | SKM | 58.38(15.14) | 41.85(4.36) | 44.55(9.00) | 31.85(3.04) |

Table 5: Performance Comparison Table of WCSS of four different estimation methods under 16 parameter conditions.

| | | $P = 50, K = 3$ | $P = 70, K = 3$ | $P = 35, K = 4$ | $P = 50, K = 4$ |
|---|---|---|---|---|---|
| $D = 20$ | GSAKM | 1442.402(73.45) | 1773.54(110.46) | 1407.62(75.33) | 1736.92(114.21) |
| , $q = 2$ | SEM | 1441.67(73.97) | 13346.09(16895.62) | 6326.59(10713.02) | 36077.63(10598.51) |
| | SEMK | 43180.33(6770.65) | 57418.31(4129.72) | 39901.59(3613.73) | 50698.78(4286.57) |
| | SKM | 43267.50(5633.81) | 58247.33(3992.27) | 39732.16(3780.62) | 50241.30(3408.44) |
| $D = 20$ | GSAKM | 2158.51(88.54) | 2638.46(146.10) | 2105.24(98.53) | 2562.77(146.31) |
| , $q = 3$ | SEM | 2158.32(88.84) | 4308.49(7912.15) | 2469.68(3255.00) | 25470.38(19878.75) |
| | SEMK | 47580.35(14465.06) | 72172.82(4317.75) | 49774.01(8435.70) | 63447.61(3470.00) |
| | SKM | 48539.67(13828.59) | 70424.93(4283.86) | 50914.74(6540.50) | 62880.66(3074.25) |
| $D = 40$ | GSAKM | 1438.60(73.56) | 1740.78(99.66) | 1397.53(69.7735.88) | 1690.39(101.31) |
| , $q = 2$ | SEM | 1437.82(73.06) | 43178.53(7100.72) | 11128.08(32422.03) | 131712.1(55251.18) |
| | SEMK | 163565.8(35394.89) | 228815.7(15910.89) | 160259.4(13578.53) | 198040.8(13786.17) |
| | SKM | 169878.3(24355.66) | 230161.8(18840.61) | 161230.6(13647.85) | 198084.6(13094.98) |
| $D = 40$ | GSAKM | 2151.98(88.10) | 2615.92(137.84) | 2102.31(96.65) | 10177.25(40213.96) |
| , $q = 3$ | SEM | 2151.75(88.27) | 6361.89(26366.08) | 2102.09(96.60) | 69359.46(76110.14) |
| | SEMK | 176308.6(67842.98) | 275009.4(14521.1) | 195217.3(33030.29) | 248687.7(12173.08) |
| | SKM | 184417.7(60941.88) | 279237.4(16053.05) | 197359.4(27946.47) | 247761.1(12133.02) |

