# OpenReview forum: "Gibbs Sampling with Simulated Annealing K-Means for Mixture Regression"
_ICLR.cc/2026/Conference — ICLR 2026 Conference Withdrawn Submission_

### Official Review · Reviewer_z2S4 · 2025-10-25

**Soundness:** 2
**Presentation:** 2
**Contribution:** 2
**Rating:** 4
**Confidence:** 3

**Summary:**

The paper addresses the challenge of fitting Mixtures of Multivariate Linear Regression (MMLR) models, where standard algorithms such as EM and K-means often converge to local optima due to the problem’s NP-hardness. To overcome this limitation, the authors propose a Gibbs Sampling with Simulated Annealing K-means algorithm, which integrates the K-means framework with Gibbs sampling and a simulated annealing schedule to improve robustness to initialization and reduce the risk of poor local minima.

**Strengths:**

The authors provide some theoretical justification for the proposed method, including non-asymptotic guarantees on convergence to the global minimum of the Within-Cluster Sum of Squares (WCSS) objective. Building on this, they derive an upper bound on the estimation error of the regression coefficients and a lower bound on classification accuracy in the asymptotic regime. Additionally, the simulation results demonstrate promising empirical performance.

**Weaknesses:**

1. The highest concern is the insufficient experimentation. The experimental section is limited to simulation studies. The paper would be significantly strengthened by including real-data experiments and qualitative analyses to demonstrate the method’s practical utility and robustness.

2. Although the method is described as being based on Gibbs sampling, which typically involves sampling from multivariate conditional distributions, the presented algorithm does not explicitly discuss a sampling procedure, which weakens the claimed connection to Gibbs sampling. For example, there is a stopping criteria to check if u converges in the algorithm, and the post-processing steps are also not clearly discussed.

3.   Some notational usage is redundant and confusing. For example, Equation (1) assumes that X follows a normal distribution with zero mean and covariance \Sigma, which is unusual since X represents fixed predictors in regression settings. Moreover, the variable \Sigma is introduced but never discussed thereafter.

4.  The proposed method combines existing, well-established techniques, Gibbs sampling, simulated annealing, and K-means without introducing a fundamentally new theoretical or algorithmic contribution. The approach reads more as a hybrid or ensemble of known methods rather than a novel framework.

5. Minor presentation issues: Some figure legends are too small to be easily readable (e.g., Figure 1), which hinders interpretability.

**Questions:**

Please see the section of weaknesses.

---

### Official Review · Reviewer_fCdh · 2025-10-30

**Soundness:** 2
**Presentation:** 2
**Contribution:** 2
**Rating:** 2
**Confidence:** 4

**Summary:**

This paper proposes a method for fitting Mixtures of Multivariate Linear Regression (MMLR) by combining Gibbs sampling with a simulated-annealing K-means algorithm. The goal is to improve robustness to initialization and avoid poor local minima that commonly affect EM and standard K-means. The authors claim non-asymptotic convergence—under a logarithmic cooling schedule—to a neighborhood of the global optimum of the Within-Cluster Sum of Squares (WCSS), along with asymptotic bounds on estimation error and classification accuracy. Empirical results across 16 synthetic settings suggest reduced estimation error compared to SEM, SEMK, and vanilla K-means.

**Strengths:**

The paper addresses an important and challenging problem—fitting Mixtures of Multivariate Linear Regression (MMLR) in the presence of non-convexity and sensitivity to initialization. It proposes a conceptually interesting hybrid approach that combines Gibbs sampling with a simulated-annealing variant of K-means, aiming to escape poor local minima and achieve more reliable clustering and parameter estimation. The attempt to provide non-asymptotic convergence guarantees toward a neighborhood of the global optimum of the WCSS objective, along with asymptotic bounds on estimation and classification error, represents a theoretically motivated step beyond standard EM or K-means methods.

**Weaknesses:**

The paper presents an interesting idea but suffers from significant shortcomings across theoretical, empirical, and expository dimensions. Below is a structured summary of its key weaknesses.


#### **1. Theoretical Weaknesses**

- **Critical theory–implementation mismatch**:
  The convergence guarantees assume a logarithmic cooling schedule satisfying
  $$
  c_1(\log t)^{-\alpha} \le T_t \le c_2(\log t)^{-\alpha}, \quad 0 < \alpha < 1.
  $$
  However, the implemented algorithm uses a modified, adaptive schedule
  $$
  T_t = T\big(\log(t_0 + t) - t_1\big)^{-\alpha},
  $$
  with dynamic rescaling of $T$ based on the running objective minimum. **No justification** is provided that this schedule satisfies the required upper and lower envelopes. Consequently, the stated theoretical guarantees **do not apply to the actual algorithm used in experiments**.

- **Incomplete or incorrect proofs**:
  - In *Lemma 3.1*, the text claims to derive an **upper bound** on $\mathbb{E}(t_k^2)$ (around line 646), but the argument yields a **lower bound**. As noted earlier (line 643), both bounds are necessary. The derivation is therefore insufficient.
  - In *Theorem 3.2*, the final misclassification probability bound (*lines 723–737*) is **inconsistent** with the version stated in the main text (*lines 284–287*), and missing intermediate steps prevent verification.

- **Unsubstantiated interpretive claims**:
  Statements such as “accurate recovery” or “misclassification decays to zero” are not explicitly linked—at the point of use—to a specific theorem or its assumptions. This overstates the scope of the theoretical results and weakens logical traceability.

- **Neglect of small component weights**:
  The analysis does not address how very small mixing proportions $p_k$ affect estimation error or convergence, despite their known role in causing instability in mixture models. Although a ridge penalty is introduced, its theoretical justification in low-$p_k$ regimes is absent.

- **Unanalyzed heuristic components**:
  The final “polish” phase—setting $T \to 0$ at the end—is described as practical but its impact on the validity of convergence guarantees is never examined. This step may violate cooling schedule assumptions.

---

#### **2. Empirical Weaknesses**

- **Synthetic-only evaluation with contradictory claims**:
  The discussion states results are based on “synthetic and real data,” yet the reproducibility statement and all figures confirm **only simulated data** were used. This inconsistency undermines credibility and external validity.

- **Lack of statistical rigor**:
  Results across 16 settings are summarized in a **single box plot** with **no confidence intervals, effect sizes, or statistical significance tests**, despite strong claims of “high-probability” accuracy. This precludes meaningful interpretation of performance gains.

- **Incomplete performance metrics**:
  The evaluation reports only **estimation error**, omitting essential diagnostics such as:
  - Classification accuracy or misclassification rate,
  - WCSS optimization trajectories,
  - Sensitivity to separation $D$, noise level, and sample size.

- **Unfair or undocumented baseline comparisons**:
  Initialization strategies, number of restarts, and tuning budgets for SEM, SEMK, and standard K-means are **not described**. It is unclear whether baselines received comparable global-search aids (e.g., K-means++ seeding or multi-start), making comparisons unreliable.

- **No ablation or sensitivity analysis**:
  The method fixes $\alpha = 0.99$, uses only **10 random initializations**, and introduces extra hyperparameters $(t_0, t_1, \kappa)$ with adaptive rescaling—yet provides **no ablation studies** for:
  - Cooling parameters ($\alpha, t_0, t_1$),
  - Number of restarts,
  - The final $T \to 0$ “polish” phase,
  - Ridge strength $\kappa$.
  This omission prevents assessment of robustness, reproducibility, or component-wise contributions.

- **Misattributed robustness to initialization**:
  Performance gains are partly attributed to running 10 independent trials (*lines 362–364*), suggesting robustness stems from **multiple restarts**, not the algorithm’s intrinsic dynamics. A direct test—e.g., varying initialization quality while fixing restart count—is needed to validate the core claim.

- **Overly favorable simulation conditions**:
  The number of components $K$ is assumed known, and inter-cluster separation $D$ is set to large values. Performance under **moderate or small $D$** (e.g., $D = 5$ or $10$) is not explored, nor is there guidance on how $K$ or $D$ would be selected in practice, limiting real-world applicability.

---

#### **3. Presentation and Technical Weaknesses**

- **Notational and typographical errors**:
  - *Model (1)* should be $Y = B_U^\top X + \epsilon$.
  - *Equation (3)* uses $\hat{u}_i = k$ instead of $u_i = k$.
  - *Line 172*: Objective should be **maximization**, not minimization.
  - *Proof of Lemma 3.1*:
    - Line 584: $\sum_{k=1}^n p_k$ → $\sum_{k=1}^K p_k$;
    - Line 591: Redundant “min” should be removed.
  - *Lines 638–641*: Missing expectation operator in Jensen’s inequality.
  - *Line 662*: References minimum eigenvalue but should cite **maximum**.
  - *Line 746*: Typo (“For \` X”).
  - *Lines 878–879*: Misplaced indexing in $\Pi^{(t)}$.

- **Ambiguous or undefined notation**:
  Constants (e.g., $c$, $C'$) and symbols appear without definition; dimensional notation is sometimes ambiguous.

- **Imprecise language**:
  Vague phrases like “high probability,” “fast convergence,” and “update law” are used without quantification or standard terminology (e.g., “update rule”). Claims lack reference to explicit rates (e.g., $\exp\{-C(\log t)^\alpha\}$), reducing scientific precision.

---

These weaknesses collectively undermine the paper’s theoretical validity, empirical credibility, and reproducibility. Substantial revisions—particularly in aligning theory with implementation, tightening proofs, and strengthening experimental design—are required before the work can be considered for publication.

**Questions:**

To address the major concerns raised in review, the authors should clarify or resolve the following points—ideally in a rebuttal or revision:

---

#### **Theoretical Questions**

1. **Cooling schedule validity**:
   The theoretical analysis assumes a logarithmic schedule satisfying
   $$
   c_1(\log t)^{-\alpha} \le T_t \le c_2(\log t)^{-\alpha}, \quad 0 < \alpha < 1.
   $$
   However, the implemented schedule is
   $$
   T_t = T\big(\log(t_0 + t) - t_1\big)^{-\alpha},
   $$
   with $T$ dynamically rescaled based on the running objective minimum.
   **Does this modified schedule satisfy the required upper and lower envelopes?** If yes, please provide a formal justification or proof.

2. **Final “polish” phase ($T \to 0$)**:
   This step is described as practical but is not covered by the theoretical analysis.
   **Does setting $T = 0$ at the end preserve the convergence guarantees?** If not, how should the theoretical claims be qualified?

3. **Behavior under small component weights**:
   The bounds do not account for very small mixing proportions $p_k$ (e.g., $< 0.01$), which are known to cause instability in mixture models.
   **How does the ridge penalty mitigate this issue, and can its effect be reflected in the error bounds?**

4. **Inconsistency in Theorem 3.2**:
   The misclassification probability bound stated in the main text (lines 284–287) differs from the one derived in the proof (lines 723–737).
   **Can you reconcile these two expressions and provide the missing intermediate steps?**

---

#### **Empirical Questions**

5. **Real-data claim**:
   The discussion states results are based on “synthetic and real data,” but all experiments appear synthetic.
   **Were real or semi-synthetic datasets used?** If so, please identify them and share code/results. If not, please correct the text.

6. **Fairness of baseline comparisons**:
   Initialization strategies, number of restarts, and tuning budgets for SEM, SEMK, and K-means are not specified.
   **Did baselines receive the same number of random starts (e.g., 10) and comparable seeding (e.g., K-means++)?** Please detail the experimental protocol for all methods.

7. **Source of robustness**:
   Performance gains may stem from multiple restarts rather than the annealing dynamics.
   **Can you isolate the algorithm’s intrinsic robustness?** For example, fix the number of restarts to 1 and compare performance under poor vs. good initializations.

8. **Sensitivity to cluster separation**:
   Experiments use large inter-cluster separation $D$.
   **How does performance degrade for moderate or small $D$ (e.g., $D = 5$ or $10$)?** Please include such results to assess practical relevance.

9. **Ablation studies**:
   The method introduces several hyperparameters ($\alpha$, $t_0$, $t_1$, $\kappa$) and an adaptive schedule.
   **What is the marginal contribution of each component?** Please provide ablation studies for key design choices.

---

#### **Presentation and Clarity**

10. **Notation and typos**:
    Several errors were noted (e.g., Model (1), Equation (3), eigenvalue reference, missing expectation in Jensen’s inequality).
    **Will these be corrected in revision?** Additionally, will all constants (e.g., $c$, $C'$) be defined on first use?

11. **Precision of claims**:
    Statements like “accurate recovery” or “misclassification decays to zero” are not explicitly tied to theorems.
    **Can you revise such claims to cite the specific result and its assumptions at the point of use?**

---

### Official Review · Reviewer_LXRL · 2025-10-31

**Soundness:** 2
**Presentation:** 2
**Contribution:** 2
**Rating:** 2
**Confidence:** 3

**Summary:**

The authors proposed a novel Gibbs sampling-enhanced simulated annealing K-means clustering algorithm for mixed multivariate linear regression models. They gave theoretical results that shows under mild assumptions, the global minimizer of the K-means
objective can recover the true regression matrix with true assignments with high probability, under both asymptotic and non-asymptotic regimes.

**Strengths:**

The proposed methods are conceptually appealing and demonstrate strong empirical performance. The simulation studies are comprehensive and show clear advantages over several competing approaches for mixed multivariate linear regression models, suggesting the effectiveness and robustness of the proposed algorithm.
In addition, the paper provides a thorough and well-structured literature review, which not only positions the work within the context of existing methods but also clearly highlights the research gap that motivates the proposed approach. This thoughtful connection between prior studies and the new contribution enhances the readability and overall value of the paper.

**Weaknesses:**

1. Unclear notations and confusing equations. I may list a few.
- In equation (1), the meaning of $u$ and the symbol $\perp$ are unclear. Does $\perp$ indicate statistical independence between $X$ and $u$, or geometric orthogonality?
- The relationship among $u$, $U$, and $\mathcal{U}$ should be clearly defined.
- In equation (2), the summation index involves $u_i$ and $k$, but the term $\|y_i - x_i B_k\|_2^2$ does not depend on $u_i$; similar confusion arises in equation (3).
- In equation (4), the parameter $\kappa$ appears without any prior definition or explanation.


2. The legitimacy and implications of Assumptions 1--3 should be discussed in more depth. For instance, how restrictive are they, and are they commonly used in mixture regression or similar models?


3. About the mathematical rigor and notation consistency.
Several mathematical expressions are imprecise or potentially misleading. For example, in Theorem 3.3, the $C'$ appears to depend on $\Sigma$, which contradicts the usual convention that we use $C'$ to denote some constant, which should be independent of model parameters.


Overall, these issues compromise the mathematical soundness and clarity of the paper. The authors are encouraged to carefully revise the notations, provide explicit definitions, and ensure the rigor and consistency of all mathematical statements.

**Questions:**

See Weaknesses

---

### Official Review · Reviewer_E537 · 2025-11-05

**Soundness:** 3
**Presentation:** 3
**Contribution:** 2
**Rating:** 2
**Confidence:** 3

**Summary:**

This article deals with the problem of the estimation of the
parameters of a mixture of linear regression models and thus with the
underlying clustering problem for the affectation of each point to a
mixture component. Contrary to most of existing methods, the
contribution provides a global optimum of the clustering cost
function. This is made possible by leveraging simulated annealing and
Gibbs sampling. Authors provides theoretical guaranties for the
clustering, a theoretical analysis of the resulting mixture parameters
and an experimental analysis on synthetic datasets.

**Strengths:**

Looking for the optimum solution of a k-means clustering problem is of
primary importance. The proposed approach provides theoretical
guaranties on the quality of the solution.

**Weaknesses:**

I feel the introduction is too generic, I was surprised to see so much
references to introductory courses (James et al., 2013), (Härdle &
Simar, 2007; Hastie, 2009) in the begining. The introduction is
supposed to put the contributions in a broader context, but something
more specific would probably be more interesting for the reader.

## About the experiments

Experiments are rather disappointing. First the section "4 Simulation
Studies" seems to be completely disorganized. The first sentence "This
section presents a comprehensive empirical evaluation..." seems
misleading, I do not see in which manner the work can be seen as
comprehensive.

The temperatures cheduling scheme should be an integral part of the
method and not presentend as an experimental detail.

"All four algorithms —-SEM, SEMK, SKM, and GSAKM —- undergo a rigorous
evaluation" - ok but where is the evaluation ? No comments or
conclusion in the text and Figure 1 (which is not reference in the
text !) is barely readable not sufficient by itself.

The title of the subsection "Simulation results" is misleading since
it's not about showin results and only describing a part of the
evaluation protocol.

The two final paragraphes are very strange. It looks like a conclusion
or a part of an introduction, but not something to put into a
experimental section. It may be a copy-paste mistake or a LLM-mess.

## Misc remarks

- Line 191: typo, "we seem"
- Line 246: minimizeS

**Questions:**

Line 151, please define κ. I guess it's the regularization paremeter ?

I do not understand why the regularization term vanishes when
T→0. Could you please elaborate ?

Line 384, I do not understand the "K=K=3" and "K=K=4".

Could you give some insight in the main text about the mains ideas of
the proofs ? I understand the space constraints, but I think it
deserves to be shown.

Theorem 3.4 shows that the output is robust to the
initialization. However you state yourself that you use in practice
kmeans-++. Could you detail the influence of the initialization on the
convergence rate ? At least experimentally and if possible
theoretically.

Subsection 4.2 is misleadingly titled "Simulation results" but
contains only comments on the experimental setups. Could you please
comment on the Figure 1 ? It is completely unreadable per se and not
referenced anywhere in the text. This is not acceptable.

Could you justify your experimental choices ? It is not
"comprehensive" at all (is it a standard LLM sentence ?...) so you
should explain why your synthectic dataset is interesting.

About Figure 1, the classification error is not shown. Could you add
something about it ? (Or remove the text about it if it's not
interesting for the reader)

---

### Note · Authors · 2025-12-02

I have read and agree with the venue's withdrawal policy on behalf of myself and my co-authors.